# CASCADIA: AN EFFICIENT CASCADE SERVING SYSTEM FOR LARGE LANGUAGE MODELS

**Youhe Jiang**[1*]**, Fangcheng Fu**[2*]**, Wanru Zhao**[3*]**, Stephan Rabanser**[4]**, Jintao Zhang**[5]**, Nicholas D. Lane**[3]**, Binhang Yuan**[1]

1. The Hong Kong University of Science and Technology
2. Shanghai Jiaotong University
3. University of Cambridge
4. Princeton University
5. Tsinghua University

```
youhejiang@gmail.com, ccchengff@sjtu.edu.cn
rabanser@princeton.edu, zhang-jt24@mails.tsinghua.edu.cn
{wz341, ndl32}@cam.ac.uk, biyuan@ust.hk
```

## ABSTRACT

Recent advances in large language models (LLMs) have intensified the need to deliver both rapid responses and high-quality outputs. More powerful models yield better results but incur higher inference latency, whereas smaller models are faster yet less capable. Recent work proposes balancing this latency–quality trade-off using model cascades, which route simpler queries to smaller models and more complex ones to larger models. However, enabling efficient cascade serving remains challenging. Current frameworks lack effective mechanisms for handling (i) the huge and varying resource demands of different LLMs, (ii) the inherent heterogeneity of LLM workloads, and (iii) the co-optimization of system deployment and routing strategy. Motivated by these observations, we introduce CASCADIA, a novel cascade serving framework designed explicitly to schedule request routing and deploy model cascades for fast, quality-preserving LLM serving. CASCADIA employs a bi-level optimization method: at the deployment level, it uses a mixed-integer linear program to select resource allocations and parallelism strategies based on LLM information and workload characteristics; at the routing level, it applies a Chebyshev-guided method to iteratively co-optimize the routing strategy and the system deployment produced by the deployment level. Our extensive evaluation on diverse workload traces and different model cascades (DeepSeek and the Llama series) demonstrates that CASCADIA significantly outperforms both single-model deployments and the state-of-the-art cascade serving baseline, achieving up to $4\times$ ($2.3\times$ on average) tighter latency SLOs and up to $5\times$ ($2.4\times$ on average) higher throughput while maintaining target answer quality.

## 1 INTRODUCTION

Large language models (LLMs) such as DeepSeek-R1 (Guo et al., 2025), OpenAI o3 (OpenAI, 2025), Claude (Anthropic, 2024), Gemini (Reid et al., 2024) and Llama-3 (Dubey et al., 2024) have demonstrated outstanding performance across a wide range of real-world applications (e.g., chatbots, healthcare and education) (Jeon & Lee, 2023; Peng et al., 2023; GitHub, 2024), largely influence human lives. However, serving LLMs can be costly (Jiang et al., 2024; 2025b; Miao et al., 2024b), since significant computational resources (e.g., GPUs) are required to meet certain service demands, such as meeting certain latency deadlines (i.e., SLO attainment—the proportion of requests served within a specified response-time target) and generation throughput. In this paper, we explore an alternative solution that strategically utilizes model cascades to better balance the response latency and quality trade-offs inherent in LLM serving.

---

[*] represents equal contribution. Correspondence to Binhang Yuan.

Cascade model serving refers to a serving architecture where multiple models of varying sizes and capabilities are arranged in a sequential pipeline, creating a hierarchy of models that process requests with increasing levels of sophistication (Aggarwal et al., 2024; Chen et al.; Kossmann et al., 2024; Kolawole et al.; Lebovitz et al., 2023; Streeter, 2018). As shown in Figure 1, larger models typically provide higher response quality but also incur greater latency, which in turn leads to increased energy consumption and compute usage (Samsi et al., 2023). In this approach, incoming requests are initially handled by smaller, computationally efficient models that can rapidly process simpler requests. Only when these lightweight models determine that a request exceeds their capabilities or requires higher-quality responses does the system escalate the request to larger, more powerful models in the cascade. This progressive delegation mechanism enables service providers to optimize system performance by matching request complexity with appropriate model capacity, thereby significantly reducing computational costs while maintaining high-quality responses for complex request. Several recent studies have focused on optimizing LLM serving using model cascades (Chen et al.; Aggarwal et al., 2024; Kossmann et al., 2024; Gupta et al.; Narasimhan et al., 2024).

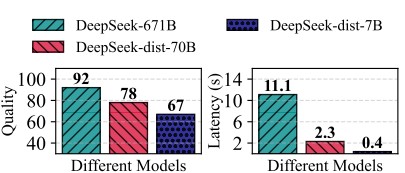

Figure 1: Average response quality and latencies of different DeepSeek models. Quality is judged by GPT-4o using the LLM-as-a-Judge framework (Zheng et al., 2023).

The cascade model serving architecture, which adaptively routes simpler and more complex requests to smaller and larger models, respectively, presents significant opportunities for optimizing the cost-efficiency of LLM serving. In this work, we focus specifically on the setting where service providers host and manage every model in the cascade themselves. However, effectively adapting this paradigm to LLM scenarios is much harder to implement than to propose, as we enumerate below:

- **Model heterogeneity.** LLMs require large amounts of compute and memory, and different models have varying resource demands for efficient serving (Duan et al., 2024). With a fixed resource pool, suboptimal allocation across models in the cascade can degrade overall serving efficiency.

- **Workload heterogeneity.** LLM workloads exhibit considerable heterogeneity (Sun et al., 2024; Zheng et al.; Zhao et al.). Models within the cascade often face incoming requests with varying characteristics (e.g., input/output lengths, arrival rates) and favor different deployment strategies (e.g., replication, parallel configuration), further adding complexity to optimal system deployment.

- **Cascade-aware load balancing.** The request routing strategy directly impacts the system load of each model in the cascade. For instance, if more requests are routed to a particular model, its load increases; the resource allocation and deployment strategy for that model should then be adjusted to balance loads across all models. Consequently, the deployment of multiple models must be co-optimized with the routing strategy to manage load across the cascade.

In order to overcome these challenges, we propose CASCADIA, a novel cascade serving system that is optimized for LLM characteristics and that co-optimizes the deployment of multiple models in the cascade together with the request routing strategy. Our contributions are as follows:

- **Contribution 1.** We formulate cascade serving—covering system deployment and request routing—as a constrained optimization problem. To solve it, we propose a bi-level approach that jointly optimizes deployment and routing. The *deployment* level uses mixed-integer linear programming (MILP) to determine the optimal deployment plan given a routing strategy, while the *routing* level applies a Chebyshev-guided method to optimize routing, balancing latency and quality.

- **Contribution 2.** We implement CASCADIA, an efficient cascade serving system tailored to LLMs. CASCADIA enables an adaptive model cascade paradigm that allocates resources and routes requests across a hierarchy of model sizes (e.g., small, medium, and large), thereby balancing response latency and output quality. Within each cascade stage, CASCADIA supports various parallelism strategies (e.g., tensor and pipeline parallelism), which allows it to automatically select the optimal strategy based on model size, incoming workload, and routing decisions.

- **Contribution 3.** We empirically evaluate CASCADIA by comparing it to both single-model and existing cascade serving systems across a variety of scenarios, including diverse workload traces (e.g., coding and mathematics), different model cascades (DeepSeek and the Llama series), and multiple evaluation metrics (SLO attainment and throughput). The results show that, compared with state-of-the-art non-cascade and cascade solutions, CASCADIA achieves up to 4× lower latency deadlines (2.3× on average) and boosts system throughput by up to 5× (2.4× on average).

## 2 PRELIMINARY AND RELATED WORK

**LLM inference phases and workload heterogeneity.** There are two phases within LLM inference: *prefill* and *decoding*. During the prefill phase, the model processes the input prompt to compute the key-value (KV) cache and generates the first token in a single step. In contrast, the decoding phase uses the last generated token and the KV cache as inputs to generate subsequent tokens in a token-by-token manner. Generally, the prefill phase is compute-bound, while the decoding phase is memory-bound (Patel et al., 2024; Zhong et al., 2024; Agrawal et al., 2024). LLM inference workloads exhibit heterogeneity in input, output token lengths and request arrival rate, which is called *workload heterogeneity*. For instance, conversation workloads (short input and long output lengths) typically require more memory resources to handle the memory-bound decoding phase, while coding workloads (long input and short output lengths) demand more compute resources to manage the compute-bound prefill phase. Therefore, appropriately allocating resources based on workload demands is critical for optimal performance (Zhao et al., 2024; Jiang et al., 2025a).

**Cascade model inference.** Current LLMs come in various sizes and configurations, offering a broad spectrum of choices. Effectively leveraging this diversity can balance trade-offs between response latency and quality during inference. Recent efforts propose cascade model inference to utilize models of differing complexities (Dekoninck et al., 2025; Narasimhan et al., 2025). In such architectures, an input prompt is processed through increasingly complex models, using threshold-based routing that stops computation once a cheaper model produces a confident enough answer. For instance, FrugalGPT (Chen et al.) employs a dynamic LLM cascade strategy that routes queries through progressively stronger models (e.g., GPT-3.5 → GPT-4) based on real-time difficulty estimation, optimizing cost-efficiency without sacrificing accuracy. Similarly, AutoMix (Aggarwal et al., 2024) uses intelligent layer-wise token routing to dynamically allocate computation based on input difficulty. CascadeServe (Kossmann et al., 2024) automates and optimizes end-to-end inference with cascades, adjusting model deployment and request routing based on real-time system loads. However, existing systems overlook key LLM-specific workload characteristics and neglect the importance of co-optimizing system deployment with request routing (i.e., system-algorithm co-design).

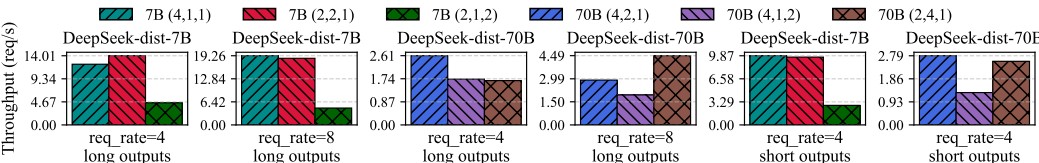

Figure 2: Benchmarked performance of different parallelism strategies across different workloads and model sizes. Long and short outputs represent two different workloads with average output sequence length to be 512 and 1024; the three-element array represents the DP, TP, and PP degrees.

**Limitations of existing cascade serving systems.** We summarize the limitations of existing cascade serving systems: (**i**) Ineffective resource allocation for different model types within a cascade. Different model types have distinct memory and computation resource needs. For example, DeepSeek-671B typically requires more allocated resources than DeepSeek-dist-70B due to its larger memory and computational demands. Current systems ignore the importance of adjusting resource allocation according to the needs of different model types, leading to unbalanced system loads. (**ii**) Inadequate adaptation of parallelism strategies to varying workloads and model sizes. The optimal parallelism strategies vary across different workloads (e.g., different input and output request sequence lengths and request arrival rates) and model sizes. As shown in Figure 2, choosing the optimal parallelism strategy can achieve up to 3× higher system throughput. Current systems do not optimize parallelism strategies according to specific workload and model size, resulting in degraded overall system performance. (**iii**) Insufficient co-optimization between system deployment and routing strategy. The routing strategy decides the request portion processed by each model type within a cascade, which in turn determines the system loads for different model types. Existing systems neglect to adapt system deployment configurations based on routing outcomes, resulting in suboptimal resource usage. To address these challenges, a cascade serving system tailored for LLMs is necessary. Such a system must optimize end-to-end performance and ensure stringent SLO adherence.

# 3 SCHEDULING ALGORITHM IN CASCADIA

## 3.1 PROBLEM FORMULATION

To optimize the cascade serving system under different LLM workloads and user-specific requirements (e.g., system response quality requirements), the scheduling algorithm should determine two essential components: (**i**) *The model deployment plan*, which specifies the resource allocations and parallelism strategies for multiple model types (e.g., small, medium, large) within the cascade to minimize the system response latency (e.g., p95 latency—the response time threshold below which 95% of all requests complete); and (**ii**) *the routing strategy*, which balances the trade-off between system response latency and quality to decide the appropriate model path for each incoming request. We term a solution addressing these two components as a *cascade plan*.

Note that the routing strategy determines the request distribution over different model types, which in turn dictates the optimal model deployment plan, while the model deployment plan defines the system response latency that feeds back into the routing decision. Given the interdependent and exponentially large search space, determining the optimal cascade plan is an NP-hard problem. To solve this problem, we adopt a bi-level optimization method that enables system–algorithm co-design, which is shown in Algorithm 1, and can be summarized as:

---
**Algorithm 1: Bi-level Scheduling Workflow**

**Require:** $\theta_0$: initial routing strategy; $\theta$: routing strategy; $q_{\min}$: quality requirement; $\tilde{\mathcal{I}}$: subsampled input workload; $\mathcal{W}$: workload distribution; $Q$: system response quality; $N$: resource limit; $\mathcal{D}$: deployment plan; $L$: system response latency; $J$: latency-quality score; $K$: consecutive stable iterations to break

**Ensure:** final routing strategy $\theta$ and deployment $\mathcal{D}$
1:   $\theta \leftarrow \theta_0$ /* $\theta_0$ detailed in §3.3 */
2:   **while** true **do**
3:     $(\mathcal{W}, Q) \leftarrow$ derived [1] from $(\theta, \tilde{\mathcal{I}})$
4:     /* Optimize deployment (§3.2) */
5:     $(\mathcal{D}, L) \leftarrow$ DeploymentSolver$(\mathcal{W}, N)$
6:     /* Optimize routing strategy (§3.3) */
7:     $(\theta, J) \leftarrow$ RoutingSolver$(L, Q, q_{\min})$
8:     /* Terminate upon convergence */
9:     **if** $J$ is stable for $K$ iters **then**
10:       **break**
11:   **return** $(\theta, \mathcal{D})$

---

- **MILP-based deployment solver**: Given the routing strategy, the deployment solver (§3.2) employs an mixed-integer linear programming (MILP) formulation to capture system resource constraints and compute the optimal deployment plan that minimizes system response latency.

- **Chebyshev-guided routing solver**: Based on the system response latency generated from the deployment solver and the user-specific quality requirement, the routing solver (§3.3) applies a Chebyshev-guided method to find the optimal routing strategy that optimizes system response latency with respect to the quality requirement.

## 3.2 MILP-BASED DEPLOYMENT SOLVER

As shown in Algorithm 1, the routing strategy (obtained from routing solver) determines how many requests should be routed to each model in the cascade, thus determining the workload distribution among models. Given the **workload distribution** and **resource limit**, the deployment solver aims to determine the optimal **deployment plan**, which includes the resource allocation and parallelism strategies for models within cascades. An example deployment plan is shown in Figure 3.

Assume a total of $N$ GPUs serve a model cascade with $C$ model types, $\{c_1, c_2, \ldots, c_C\}$, where $c_i$ denotes the $i$-th model type. The incoming workload information is denoted as $\mathcal{W} = \{w_1, w_2, \ldots, w_C\}$, where each $w_i$ includes the distributions of input/output sequence lengths and the request arrival rate for the $i$-th model type. We use $\mathcal{F} = \{f_1, f_2, \ldots, f_C\}$ to denote the number of GPUs allocated per model, the total allocation must not exceed the resource limit, i.e., $\sum_{i=1}^{C} f_i \leq N$. Given this setup, our deployment solver (**i**) determines the parallelism strategy for each specific resource allocation $f_i$, and (**ii**) uses an MILP to optimize the overall resource allocation $\mathcal{F}$.

**Parallelism strategy search.** Given the workload information $w_i$ and a specific resource allocation $f_i$, this optimization determines the optimal parallelism strategy and computes the corresponding system response latency $l_i$ for the model type $i$. CASCADIA provides three forms of parallelism: data parallelism (i.e., model replication, DP) (Li et al., 2023), tensor model parallelism (TP) (Shoeybi et al., 2019), and pipeline parallelism (PP) (Huang et al., 2019). Denoting the degrees of data, tensor,

---

[1]Given $\theta$ and $\tilde{\mathcal{I}}$, $\mathcal{W}$ is derived by aggregating per-model routed requests (including arrival rates and sequence statistics), while $Q$ is derived by aggregating quality scores of accepted outputs across all models (Chen et al.).

and pipeline parallelism for the model type by dp, tp, and pp, any feasible parallelism strategy must satisfy the following resource constraint: $(\sum_{j=1}^{\text{dp}_i} \text{tp}_{i,j} \times \text{pp}_{i,j}) \leq f_i$, i.e., one model type can be replicate into multiple replicas, each replica can have varied tensor and pipeline parallelism degrees, as shown in Figure 3, the summation of different parallelism degrees should be less or equal than the total number of GPUs assigned. Based on the workload information $w_i$ and the resource allocation $f_i$, we iterate over all feasible parallelism combinations to select the strategy that minimizes the response latency $l_i$ for the model type $i$. The latency $l_i$ is computed using the simulator $\mathbf{Sim}(\cdot)$ as $l_i = \mathbf{Sim}(w_i, f_i)$ [2]. Note that the parallelism strategy optimization can be precomputed for all possible resource allocations $f$ to provide latency lookup tables for the MILP formulation.

**MILP formulation for resource allocation optimization.** Our MILP problem formulation aims to minimize the maximum system response latency among all model types in the cascade. Let $L$ denote the maximum latency across all model types. We discretize the GPU allocations into candidate values $f \in \{1, 2, \ldots, N\}$. For each model type $i$ and candidate allocation $f$, we use the precomputed latency table from the

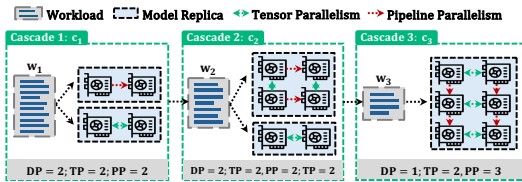

Figure 3: Illustration of a model deployment plan.

parallelism strategy optimization to obtain $l_i(f)$. We then introduce binary assignment variables $x_{i,f}$, where $x_{i,f} = 1$ if model type $i$ is assigned $f$ GPUs and $x_{i,f} = 0$ otherwise, for all $i \in \{1, \ldots, C\}$ and feasible $f$. The constraints of our MILP include: (**i**) For each model type $i$, exactly one GPU allocation $f$ must be selected, i.e., $\sum_{f=1}^{N} x_{i,f} = 1, \forall i = 1, \ldots, C$; (**ii**) the total number of GPUs assigned across all model types should be equal to the available GPUs $N$, i.e., $\sum_{i=1}^{C} \sum_{f=1}^{N} f x_{i,f} = N$; and (**iii**) the maximum latency $L$ must be at least as large as the latency $l_i(f)$ corresponding to each selected allocation, i.e., $L \geq \sum_{f=1}^{N} l_i(f) x_{i,f}, \forall i = 1, \ldots, C$. We explicitly enforce variable domains and integrality constraints as follows: $x_{i,f} \in \{0, 1\}, \forall i, f$ and $L \geq 0$. If certain GPU allocations $f$ are infeasible for specific model types—such as when the total memory of the allocated $f$ GPUs is less than the minimum memory required by the model type—we explicitly set $x_{i,f} = 0$ for these allocation pairs. Our objective is to minimize the maximum system response latency $L$, which serves as the input for the routing layer optimization.

### 3.3 CHEBYSHEV-GUIDED ROUTING SOLVER

As shown in Algorithm 1, the deployment plan (obtained from the deployment solver) determines the system response latency. Given the **system response latency** and **quality requirement**, the routing solver aims to optimize the **routing strategy** (i.e., co-optimize system latency and quality).

**Thresholds tuning and request routing.** We adopt the threshold-based cascade routing workflow consistent with prior works (Aggarwal et al., 2024; Chen et al.) (Figure 4). Initially, every incoming request is sent to the first (smallest) model type $c_1$ in the cascade. A judger then evaluates the quality of the output responses from model types $c_1$ to $c_{C-1}$, and a set of thresholds $\mathcal{H} = \{h_1, h_2, \ldots, h_{C-1}\}$ is defined to decide whether the requests at each model type should be accepted or forwarded to the next model type. In this framework, the routing strategy $\theta$ is directly determined by the thresholds $\mathcal{H}$, i.e., $\theta = \theta(\mathcal{H})$. Each routing strategy $\theta$ is associated with a system response latency $L(\theta)$ (determined by the deployment solver optimization) and quality $Q(\theta)$ (determined by the judger [3]).

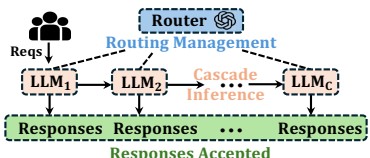

Figure 4: Threshold-based cascade routing workflow. The router determines whether a request is accepted or forwarded to the next model type based on predefined thresholds.

---

[2]We use the ETH EASL Scratchpad simulator (ETH-EASL, 2025) to estimate system p95 latency from workload and resource allocation. We show detailed simulator design (e.g., simulator inputs, batching strategy, queuing mechanism, parallelism strategy modeling) and evaluation in Appendix B.

[3]Analogous to (Chen et al.), we estimate $Q(\theta)$ by profiling a subsample of the input workload across all cascade models to obtain per-model quality score distributions. During scheduling, given any threshold vector $\mathcal{H}$ and the quality score distributions, we can determine which model's response would be accepted for each request under routing policy $\theta(\mathcal{H})$, then aggregate these final model scores to compute the overall system quality $Q(\theta)$.

Our routing solver uses a Chebyshev-guided method to optimize the routing strategy. We initialize the routing strategy $\theta_0$ as proportional routing, where the $i$-th model receives $1/i$ of requests.

**Chebyshev-guided optimization for routing strategy.** Given the routing strategy $\theta$ and user-specified quality requirement $q_{\min}$, we employ the Chebyshev-guided method (Steuer & Choo, 1983) to minimize the system response latency $L(\theta)$ with respect to $q_{\min}$. First, we define a utopia point $z_1^*$ (all requests processed by the largest model $c_C$) and nadir point $z_2^*$ (all requests processed by the smallest model $c_1$) representing the best and worst achievable system response quality. Then, for a given quality requirement $q_{\min}$, we minimize the system response latency subject to meeting the quality requirement by solving the single-objective penalty problem:

$$\arg\min_{\theta} J(\theta) = \arg\min_{\theta} \left[ L(\theta) + \mu \max\{0, \; (q_{\min} - Q(\theta))/(z_1^* - z_2^*)\} \right]$$

where $J(\theta)$ represents the latency-quality score, $\mu > 0$ is a penalty weight that enforces the quality constraint (for sufficiently large $\mu$, any minimizer of $J(\theta)$ satisfies $Q(\theta) \geq q_{\min}$), and $z_1^*$ and $z_2^*$ are used to normalize the quality shortfall so the penalty is dimensionless and well-conditioned across workloads. Note that our routing solver can also optimize system response quality under a user-specified latency requirement using a similar procedure, as detailed in Appendix C.

> **Illustrative example for Chebyshev-guided optimization.** Assume the utopia and nadir points $z_1^*$ and $z_2^*$ equal 0.95 and 0.75. The user-specific quality requirement $q_{\min}$ is 0.90 and the penalty weight $\mu$ is 100. Consider a strategy $\theta_1$ with p95 latency $L(\theta_1) = 11.0$ s and overall quality $Q(\theta_1) = 0.88$. The normalized shortfall from the requirement is $(0.90 - 0.88)/(0.95 - 0.75) = 0.02/0.20 = 0.10$, yielding $J(\theta_1) = 11.0 + 100 \times 0.10 = 21.0$. Consider another strategy $\theta_2$ with latency $L(\theta_2) = 11.4$ s and quality $Q(\theta_2) = 0.91$, which results in $J(\theta_2) = 11.4$. Strategy $\theta_2$ is preferable under this setting due to its significantly lower objective value. Additionally, a higher-quality strategy $\theta_3$ with latency $L(\theta_3) = 12.2$ s and quality $Q(\theta_3) = 0.93$ yields $J(\theta_3) = 12.2$. Although both $\theta_2$ and $\theta_3$ satisfy the quality requirement $q_{\min}$, strategy $\theta_2$ is preferable since it achieves lower latency while meeting the constraint. This example demonstrates how the Chebyshev-guided method effectively penalizes infeasible solutions while optimizing system response latency.

**Putting them together.** In our bi-level optimization framework, the routing solver (i.e., Chebyshev-guided optimization) iteratively searches for the next $\theta$, invokes deployment solver (i.e., MILP optimization) to obtain the minimized system response latency $L(\theta)$, and then minimizes the objective function (i.e., $\arg\min_{\theta} J(\theta)$). Finally, an optimal routing strategy $\theta$ is selected that guarantees a minimal system response latency while fulfilling the quality requirement.

**Impact of LLM workloads on optimal cascade plan selection.** The characteristics of incoming LLM workloads strongly influence the selection of cascade plans. This influence stems from two key factors: (**i**) Request input/output length and arrival rate affect system response latency—longer sequences or higher loads increase compute demand, necessitating plan adjustments to balance latency and quality; (**ii**) Request complexity impacts system response quality—complex requests or difficult queries require larger models, necessitating plan adjustments to maintain quality while managing latency. Therefore, our bi-level optimization framework considers both system performance (e.g., deployment solver) and algorithmic behavior (e.g., routing solver), enabling efficient, adaptive optimization across different incoming LLM workloads. Additionally, our framework incorporates a re-scheduling mechanism to handle online fluctuating workloads, as detailed and tested in §4.4.

The complete mathematical formulation for our bi-level optimization is provided in Appendix D.

# 4 EVALUATION

## 4.1 EXPERIMENTAL SETUP

**Environments.** Our experiments are conducted on 4 GPU servers, where each server is equipped with 8 NVIDIA H100-80GB GPUs. Within each server, the GPUs are connected via NVLink with a bandwidth of 400GB/s, and the servers are connected via Inifiband with a bandwidth of 200GB/s.

**Model cascade construction.** We construct a model cascade using the DeepSeek series models for CASCADIA, which are representative and popular open-source transformer models. Specifically, we use DeepSeek-dist-7B, DeepSeek-dist-70B (distilled version), and DeepSeek-671B AWQ with INT4 quantized weights (Lin et al., 2024) as three model types within our system. We employ a GPT-4o (LLM-as-a-Judge) (Zheng et al., 2023) as the judger mentioned in §3.3, which assesses the output

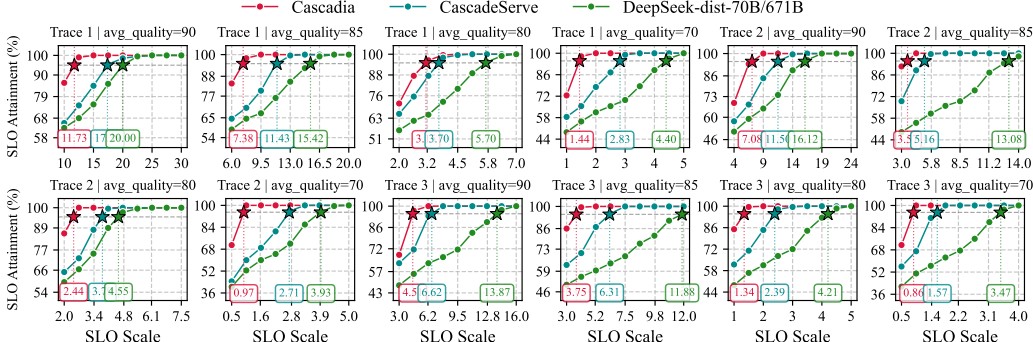

Figure 5: End-to-end SLO attainment results evaluating CASCADIA against two baseline systems. Each row corresponds to a particular LLM workload trace, and each column corresponds to a specific quality requirement. The stars indicate the 95% SLO attainment for each system.

responses of each model type within the cascade and assigns scores between 0 and 100. The judging overhead [4] is included in our experiments.

**Baselines.** We compare CASCADIA with two baselines:

- **Compare with stand-alone LLMs served by SGLang.** We compare CASCADIA against stand-alone LLMs that are directly served on SGLang (Zheng et al., 2024) under various response quality constraints (e.g., 90, 85, 80, 70) to demonstrate the effectiveness of LLM serving with model cascades. For quality requirement of 90 and 85, we choose stand-alone DeepSeek-671B for comparison, and for quality reqirement of 80 and 70, we choose stand-alone DeepSeek-dist-70B for comparison. For fair comparison, we tune the parallelism strategy using our MILP algorithm mentioned in §3.2 for each of the stand-alone model and report the best values in all experiments.

- **Compare with cascade model serving system CascadeServe (Kossmann et al., 2024).** We compare CASCADIA against an existing cascade model serving system CascadeServe. It chooses model cascade deployment plan based on system load (e.g., request arrival rate), enables model replication on hardware and adaptively dispatches incoming requests. We tune the parallelism and request routing strategies for CascadeServe based on the real-time system load and report the best values in all experiments.

**Traces.** We follow prior work to generate workload traces based on real-world data (Jiang et al., 2024; Zhong et al., 2024). Our testing traces are subsampled from MT-Bench (Zheng et al., 2023), a multi-turn conversation benchmark that contains multiple types of LLM workloads (e.g., coding, mathematics and reasoning). Each of our subsampled traces have different workload characteristics and different complexities as mentioned in §3.3.

**Evaluation metrics.** Following previous evaluation setups (Li et al., 2023; Duan et al., 2024; Agrawal et al., 2024), we evaluate system performance based on SLO attainment and system throughput. The SLO is determined empirically based on the system's average single-request processing latency, and we scale it to various multiples (SLO Scale in Figure 5) to assess performance under different levels of operational stringency. We focus on identifying the minimum SLO Scale at which the system achieves 95% SLO attainment.

## 4.2 END-TO-END EXPERIMENTAL RESULTS

**End-to-end system performance.** We evaluate the SLO attainment and throughput of CASCADIA across multiple traces and quality requirements, comparing it with two baselines. Results in Figure 5 and Figure 6 show that CASCADIA outperforms all baselines:

- CASCADIA achieves up to 4× and on average 2.8× lower latency deadlines, and up to 5× and on average 3× higher system throughput compared with stand-alone LLMs. For instance, when testing on trace 3 with an average quality requirement of 85, stand-alone DeepSeek-671B requires

---

[4]The judger takes a Q&A pair as input and outputs quality grades (1–2 tokens), resulting in significantly lower latency and cost than full request inference (on average 0.27s for a single judge). We benchmark the judge overhead in Appendix H. We also demonstrate sensitivity experiments when replacing GPT-4o with weaker judgers (e.g., GPT-4o-mini and Llama3.1-70B) in Appendix K.

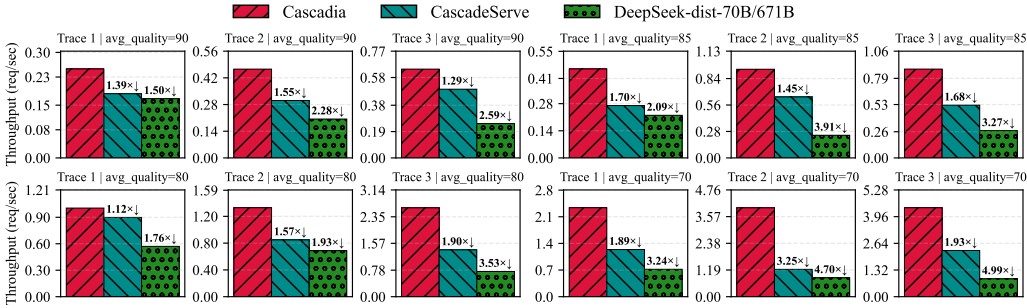

Figure 6: End-to-end throughput results evaluating CASCADIA against two baseline systems across different LLM workload traces and quality requirements.

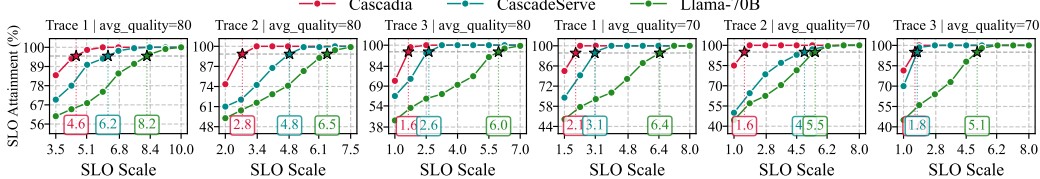

Figure 7: End-to-end SLO attainment results evaluating CASCADIA against two baselines using a Llama cascade (Llama3-8B; Llama3-70B) across LLM workload traces and quality requirements.

11.88 SLO scales to achieve 95% attainment, while CASCADIA with different model types that uses smaller models to process simpler requests only requires 3.75 SLO scales.

- CASCADIA achieves up to $2.5\times$ and on average $1.7\times$ lower latency deadlines, and up to $3.3\times$ and on average $1.7\times$ higher throughput than CascadeServe. While CascadeServe optimizes model deployment and routing based on real-time load, it overlooks LLM-specific workload characteristics (e.g., input/output lengths) and request complexity, leading to sub-optimal parallelism and routing. For example, on trace 1 with an average quality requirement of 90, CascadeServe needs 17.3 SLO scales to reach 95% SLO attainment, whereas CASCADIA requires only 11.73.

**System performance with different model cascades and serving optimizations.** We further evaluate CASCADIA using a different model cascade by replacing the DeepSeek series with the Llama series (Llama3-8B and Llama3-70B). As shown in Figure 7, CASCADIA outperforms baselines by up to $3.8\times$ and on average $2.6\times$, demonstrating strong performance across LLM cascades. We also compare CASCADIA with Sarathi-Serve (Agrawal et al., 2024), a serving system with chunked prefill optimizations. CASCADIA achieves $1.95\times$ higher performance ($1.64\times$ average), validating our approach against advanced systems with scheduling optimizations. Detailed results are in Appendix F.

**Compare with RouteLLM.** We added additional experiments comparing CASCADIA with RouteLLM, a LLM routing framework. CASCADIA achieves on average 21.3% lower SLO scale in achieving 95% SLO attainment and 18.8% higher throughput compared to RouteLLM. CASCADIA's performance advantage stems from its system-algorithm co-design, as detailed in Appendix I.

**Cost efficiency results.** In addition to performance metrics, we conducted an analysis of cost efficiency comparing CASCADIA against baselines. Our results, detailed in Appendix L, demonstrate that CASCADIA significantly reduces operational expenditure. Specifically, CASCADIA achieves an average cost reduction of 20–39% compared to CascadeServe and a 33–61% reduction compared to stand-alone model serving, confirming its economic viability.

### 4.3 CASE STUDIES ON MODEL DEPLOYMENT PLANS AND ROUTING STRATEGIES

**Case study on resource allocation and routing strategies.** We benchmarked the thresholds, processing ratios and allocated resources for different model types across different testing cases. For instance, when testing on trace 1 with an average quality requirement of 90, model types $c_1$ to $c_3$ process 100%, 94% and 50% of the total requests, and the assigned GPU numbers are 4, 8 and 20. When the quality requirement changes to 85, less requests are required to be processed by the largest model $c_3$ (from 50% to 21%), and less resources are allocated to $c_3$ accordingly (from 20 to 16). This algorithm and system co-optimization enables CASCADIA to adjust system resource allocation and request routing based on user requirements, ensuring balanced load across different

Cascadia · Uniform Parallelism Strategy · Uniform Resource Allocation

Figure 9: Ablation study on resource allocation and parallelism strategy.

model types to boost system performance. Additionally, when testing on trace 3 with an average quality requirement of 70, CASCADIA deploys a subset of model types (DeepSeek-dist-7B and -70B) to minimize the latencies required for requests processing. As shown in Figure 8, across different testing cases, CASCADIA always balances the loads among different model types to ensure optimized system performance. Table 2 in Appendix E demonstrates the thresholds, processing ratios and allocated resources for different model types across different testing cases.

**Case study on parallelism strategies.** We benchmarked the parallelism strategies for different model types across different testing cases. For example, when testing on trace 1 with an average quality requirement of 90, the optimal parallelism strategy $s_2$ for $c_2$ is (DP=2, TP=4). In this case, if we change the parallelism strategy to (DP=4, TP=2), the performance of this model type would drop by 33.7%. Additionally, when the quality requirement drops to 85, the optimal parallelism strategy $s_2$ for $c_2$ shifts to (DP=6,

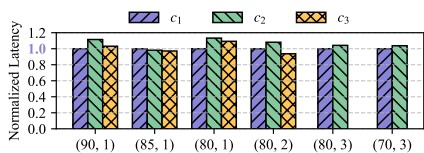

Figure 8: Benchmarked p95 latency of each model type within the cascade across different testing cases.

TP=2). This adjustment occurs because the change in quality requirements alters the LLM workloads, the request complexity routed and the resource allocated to $c_2$. Consequently, $s_2$ is updated to optimize the single model type's performance while balancing loads across all model types within the cascade. Table 3 in Appendix E presents the parallelism strategies for each model type within the cascade across different test cases.

**Ablation study.** We disable individual optimizations in CASCADIA to evaluate their impact, as shown in Figure 9: (**i**) Replacing our parallelism strategy optimization with a uniform parallelism strategy—tensor parallelism within each server and data parallelism across servers—reduces performance by up to $1.6\times$ ($1.4\times$ on average). For example, DeepSeek-7B and DeepSeek-671B requires higher degrees of data and tensor parallelism to maximize throughput and parameter sharding; a uniform approach fails to accommodate these needs. (**ii**) Replacing our resource allocation optimization with uniform resource allocation reduces performance by up to $2.1\times$ ($1.7\times$ on average). For instance, in trace 1 with an average quality requirement of 90, DeepSeek-671B was originally allocated 20 GPUs, but uniform allocation assigns only 12, causing load imbalance.

### 4.4 EFFECTIVENESS OF THE SCHEDULING ALGORITHM

**Overall scheduling process.** During scheduling, our Chebyshev-guided optimization (§3.3) explores different routing strategies to reduce response latency given a required quality. Simultaneously, our MILP-based optimization (§3.2) searches for resource allocations and parallelism strategies to balance load across model types and minimize latency. CASCADIA then selects the optimal plan—including thresholds, resource allocations, and parallelism strategies—based on quality requirements.

**Scheduling algorithm runtime and scalability.** Figure 10 shows the runtime performance of CASCADIA's scheduling algorithm, evaluated on a 12-core CPU instance. In our setup (32 GPUs), scheduling completes within 20s. For larger clusters (e.g., 80 GPUs), it finishes within one minute. These results demonstrate the algorithm's efficiency and scalability across test cases and cluster sizes. Moreover, the algorithm is highly parallelizable, as resource allocations, parallelism, and routing

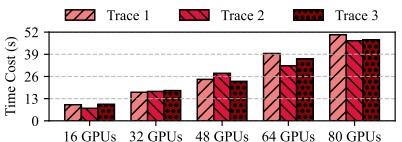

Figure 10: Algorithm running time when scaling from smaller clusters (e.g., 16 GPUs) to larger clusters (e.g., 80 GPUs).

strategies are independent—allowing execution time to scale down with more CPU cores. We added additional scheduling optimality analysis in Appendix J.

**Re-scheduling to adapt to online workload changes.** As discussed in §3.3, LLM workload characteristics (e.g., distributions of input and output lengths, request rate and complexity) significantly

affect the optimal model deployment plan and routing strategy. Thus, analogous to DistServe (Zhong et al., 2024), CASCADIA implement a re-scheduling mechanism to accommodate dynamic LLM workloads. Concretely, the system (**i**) subsample[5] and record the real-time characteristics of the incoming LLM workloads (e.g., subsample 50 requests every 5 minutes and record the workload characteristics), (**ii**) upon detecting a significant shift in workload characteristics (e.g., an increase in request arrival rate or request complexity), the scheduling algorithm is executed again, incorporating recent historical data to produce an updated deployment plan and routing strategy.

We evaluated our system against baselines under online fluctuating workloads, where the workload transitions trace 1 → trace 2 → trace 3 with segment lengths of 8, 16, and 10 minutes, evaluated at different quality constraints. As shown in Figure 11, CASCADIA consistently outperforms baseline systems, achieving up to 4.4×

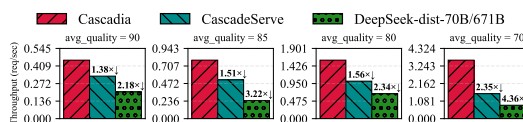

Figure 11: Throughput evaluation under fluctuating workloads.

improvement with an average of 2.2× better performance. We further demonstrate the system latency results of CASCADIA in comparison with CascadeServe and stand-alone model serving on online fluctuating workloads (see Appendix G). Despite incurring additional scheduling overhead, CASCADIA maintains superior throughput and end-to-end efficiency under fluctuating workloads by dynamically optimizing cascade plans based on real-time LLM workload characteristics.

## 5 CONCLUSION

This paper proposes CASCADIA, a cascade serving system tailored for LLMs. Its core component is a scheduling algorithm that jointly optimizes resource allocation, parallelism, and routing within the cascade system. Extensive experiments on diverse workload traces and multiple model cascades show that this co-design substantially reduces request latency and boosts system throughput compared with both single-model and existing cascade baselines, while maintaining the target answer quality.

## ACKNOWLEDGMENT

This work is supported by the HKUST startup grant R9895 from CSE; RGC-ECS project 26218024; RGC-NSFC project CRS_HKUST601/24. We thank the support from National Supercomputer Center in Guangzhou Nansha Sub-Center.

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

Table 1: Simulator accuracy across parallelism configurations on Llama3-70B model under a workload with average input and output lengths of 1600 and 16. Errors are absolute percentage errors.

| Config (DP,TP,PP) | Real (req/s) | Estimated (req/s) | Abs. % Error |
|---|---|---|---|
| (1, 4, 1) | 0.21 | 0.219 | 4.29% |
| (1, 2, 2) | 0.26 | 0.280 | 7.69% |
| (1, 1, 4) | 0.27 | 0.287 | 6.30% |
| (2, 1, 2) | 0.33 | 0.347 | 5.15% |
| (2, 2, 1) | 0.40 | 0.408 | 2.00% |
| (2, 4, 1) | 0.41 | 0.437 | 6.59% |
| (2, 2, 2) | 0.55 | 0.559 | 1.64% |

## A    EXTENDED RELATED WORK

**Parallelism strategies.** LLMs with huge memory and computational resource requirements typically rely on parallelization across multiple GPUs (Li et al., 2023). There are three prevalent forms of parallelism: data parallelism (DP, i.e., model replication), tensor parallelism (TP) (Shoeybi et al., 2019), and pipeline parallelism (PP) (Huang et al., 2019). DP replicates the model into multiple replicas, enabling parallel processing of requests. TP divides model weights and computationally intensive operations such as matrix multiplication across various GPUs, thereby splitting data scanning and computation to minimize LLM inference latency. PP divides the layers of a model into multiple stages. These stages are assigned to distinct GPUs for execution and they establish a pipeline. Only inter-layer activations are needed to be communicated between stages.

**Speculative decoding and early-exit in LLM inference.** Speculative decoding uses a lightweight draft model to generate token blocks, which a larger target model verifies—leveraging model heterogeneity to reduce computation and latency (Leviathan et al., 2023; Miao et al., 2024a; Liu et al.). Similarly, early-exit networks add decision branches at intermediate layers, enabling inference to stop early when confidence is high—cascading computation within a single model (Teerapittayanon & McDanel, 2016; Rahmath P et al., 2024). In contrast, we focus firmly on cascade model inference.

## B    SIMULATOR DESIGN AND VALIDATION

Our simulator employs a round-robin strategy for request dispatching among multiple parallel models, and a first-come first-served strategy for per-model request processing. The single-GPU processing time is based on profiled characteristics like compute TFLOPS and memory bandwidth. The simulator also considers the phase-specific characteristics of LLMs. The prefill phase is compute-bound, so its batched processing capacity is determined by the sum of the individual latencies. In contrast, the decoding phase is memory-bound, and its batched processing capability is defined by a single latency value. This distinction has been validated in several studies (e.g., DistServe (Zhong et al., 2024), Splitwise (Patel et al., 2024)).

**Inputs of the simulator.** The simulator requires three fundamental inputs: (i) the distributions of input and output sequence lengths for each model type within the cascade; (ii) the request arrival rate corresponding to each model type within the cascade; and (iii) the resource allocation designated for each model type within the cascade.

**Example.** Consider a workload distribution $\mathcal{W}$ that routes 100, 70, and 30 requests to model types 1, 2, and 3 respectively within the cascade, with corresponding GPU allocations of 2, 4, and 2 units. In this configuration, we record the distributions of input and output sequence lengths for each subset of requests (100, 70, and 30 respectively) as input files to the simulator, configure the request arrival rates and resource allocations according to the specified parameters, and execute the simulation. Subsequently, the simulator undergoes iterative execution to identify the optimal parallelism strategy based on the provided input files, request arrival rates, and resource allocation constraints.

**Batching strategy in our simulator.** The simulator's internal batching strategy is continuous batching, which iteratively batches request tokens to fully utilize the current resources. The GPU's memory limit constrains the maximum batch size for continuous batching.

**Queuing mechanism.** Our simulator maintains an individual queue for each model. Once there is free memory on the GPU (one request has finished), the model will fetch the next request in the queue for processing.

**Different parallelism.** Tensor and pipeline parallelism both split the computation workload of a single model across multiple devices. For pipeline parallelism, the simulator models communication overhead by profiling the relationship between estimated communication volume and observed latency. For tensor parallelism, the simulator assumes that each operator's computation cost ideally scales down by a factor of $1/N$ when split across $N$ GPUs, and then adjusts this ideal cost using a speed-up coefficient $K(N)$ obtained from micro-benchmarks to account for communication and synchronization overhead. All profiling is performed offline before scheduling begins.

**Simulator evaluation.** We present the accuracy of our simulator with real-time experiments in Table 1. The table presents examples of our throughput estimation for the Llama3-70B model under a workload with average input and output lengths of 1600 and 16, respectively. The notation (1,2,2) indicates a DP degree of 1, TP degree of 2, and PP degree of 2. Although the estimations are not perfectly accurate, they are sufficiently reliable (with estimation errors within 2%–7%) for selecting optimal configurations.

## C   ROUTING SOLVER IN LATENCY-CONSTRAINED CASE

The routing solver can also optimize system response quality under a user-specified latency budget by solving

$$\arg\min_{\theta} \left[ -Q(\theta) \ + \ \nu \, \frac{\max\{0, \ L(\theta) - L_{\max}\}}{z^\star_{\text{lat, max}} - z^\star_{\text{lat, min}}} \right],$$

where $z^\star_{\text{lat, min}}$ and $z^\star_{\text{lat, max}}$ are the best (minimum) and worst (maximum) achievable latencies, $L_{\max}$ is the allowable latency budget, and $\nu > 0$ scales the penalty. The same routing–deployment alternation, deployment solver, and convergence procedure are reused unchanged.

## D   COMPLETE BI-LEVEL OPTIMIZATION FORMULATION

**Problem setup and notation.** We consider a cascade with $C$ model types/stages indexed by $\{1, \ldots, C\}$ and labeled $\mathcal{C} = \{c_1, \ldots, c_C\}$, where $c_i$ denotes the $i$-th model type. The routing strategy is denoted by $\theta$, parameterized by thresholds $\mathcal{H} = \{h_1, \ldots, h_{C-1}\}$, with $\Theta$ the feasible set of routing strategies. The GPU resource allocation is $\mathcal{F} = \{f_1, \ldots, f_C\}$, where $f_i \in \mathbb{Z}_+$ is the number of GPUs assigned to model type $i$, subject to a total budget $N \in \mathbb{Z}_+$. The parallelism plan is $\mathcal{S} = \{\text{DP}_i, \text{TP}_{ij}, \text{PP}_{ij}\}_{i,j}$, where $\text{DP}_i$ denotes the number of data-parallel replicas and, for each replica $j$, $\text{TP}_{ij}$ and $\text{PP}_{ij}$ denote its tensor- and pipeline-parallel degrees. Given routing $\theta$ and deployment $(\mathcal{F}, \mathcal{S})$, the estimated p95 latency is $L(\theta, \mathcal{F}, \mathcal{S})$, and the system quality is $Q(\theta; \tilde{\mathcal{I}})$ estimated by a judger using a subsampled workload $\tilde{\mathcal{I}}$. For Chebyshev-style normalization of quality, we use quality anchors $z^\star_1$ (utopia/best achievable quality, e.g., all requests at $c_C$) and $z^\star_2$ (nadir/worst credible quality, e.g., all requests at $c_1$). A user-specified quality requirement is $q_{\min}$, and $\mu > 0$ is a penalty weight.

**Bi-level formulation.** The routing is optimized by a single scalar objective that penalizes quality shortfall, normalized by the utopia–nadir range, while the deployment is optimized under the GPU budget and parallelism feasibility:

$$\theta \in \arg\min_{\theta' \in \Theta} \left[ L(\theta', \mathcal{F}^\star, \mathcal{S}^\star) \ + \ \mu \, \max\left\{ 0, \ \frac{q_{\min} - Q(\theta'; \tilde{\mathcal{I}})}{z^\star_1 - z^\star_2} \right\} \right],$$

$$(\mathcal{F}^\star, \mathcal{S}^\star) \in \arg\min_{\mathcal{F}, \mathcal{S}} L(\theta', \mathcal{F}, \mathcal{S}) \quad \text{s.t.} \quad \sum_{i=1}^{C} f_i \leq N, \qquad \sum_{j=1}^{\text{DP}_i} \text{TP}_{ij} \text{PP}_{ij} = f_i \ \ (i=1, \ldots, C),$$

$$f_i, \ \text{DP}_i, \ \text{TP}_{ij}, \ \text{PP}_{ij} \in \mathbb{Z}_+.$$

**Tractability and solution strategy.** Because the problem couples routing, resource allocation, parallelism, heterogeneous LLM workloads, and user-specific quality requirements, a monolithic

solve is intractable. We therefore adopt a bi-level strategy: The deployment problem is solved as a MILP with latency values obtained from resource allocation and parallelism strategy optimization; the routing solver solves the Chebyshev-guided penalty problem. The two phases are executed iteratively, with the routing solver updating $\theta$ and the deployment solver resolving $(\mathcal{F}^\star, \mathcal{S}^\star)$ accordingly, and termination declared once the routing objective stabilizes under a prescribed horizon.

**Interpretation.** The bi-level problem decomposes into **routing** and **deployment** subproblems that are solved iteratively.

**Deployment solver (deployment under resource/feasibility constraints).** For a fixed routing $\theta'$, the deployment solver selects the latency-optimal deployment by choosing GPU allocations and parallelism plans subject to the budget and structural constraints:

$$(\mathcal{F}^\star, \mathcal{S}^\star) \in \underset{\mathcal{F}, \mathcal{S}}{\arg\min} \, L(\theta', \mathcal{F}, \mathcal{S}) \quad \text{s.t.} \quad \sum_{i=1}^{C} f_i \leq N, \qquad \sum_{j=1}^{\mathrm{DP}_i} \mathrm{TP}_{ij}\mathrm{PP}_{ij} = f_i \ \ (i=1,\ldots,C),$$

$$f_i, \ \mathrm{DP}_i, \ \mathrm{TP}_{ij}, \ \mathrm{PP}_{ij} \in \mathbb{Z}_+.$$

This solver captures both hardware limits (GPU budget $N$) and parallelism feasibility.

**Routing solver (routing, Chebyshev-guided optimization).** Given the current deployment $(\mathcal{F}^\star, \mathcal{S}^\star)$, the routing solver updates the routing strategy (i.e., $\theta$) by minimizing a single scalar objective that balances latency and a normalized quality shortfall:

$$\theta \in \underset{\theta' \in \Theta}{\arg\min} \left[ L(\theta', \mathcal{F}^\star, \mathcal{S}^\star) \, + \, \mu \, \max\left\{ 0, \, \frac{q_{\min} - Q(\theta'; \tilde{\mathcal{I}})}{z_1^\star - z_2^\star} \right\} \right].$$

Here, $(z_1^\star - z_2^\star)^{-1}$ provides Chebyshev (utopia–nadir) normalization for scale stability, and $\mu > 0$ sets the severity of penalizing $Q(\theta') < q_{\min}$. For sufficiently large $\mu$ (when the target is feasible), any minimizer is quality-compliant and the routing objective effectively reduces to minimizing latency among feasible routings.

**Coupling and procedure.** The routing solver's $\theta$ determines the workload distribution seen by each model type within the cascade (and hence the optimal deployment plan for the deployment solver), while the deployment solver's $(\mathcal{F}^\star, \mathcal{S}^\star)$ determines the latency used by the routing objective (and hence the optimal routing strategy for the routing solver). Alternating updates continue until the routing objective stabilizes under a prescribed termination horizon (e.g., best-so-far objective unchanged for $K$ consecutive iterations).

## E    CASE STUDIES ON MODEL DEPLOYMENT PLANS AND ROUTING STRATEGIES

**Case study on resource allocation and routing strategies.** Table 2 demonstrates the case study of thresholds, processing ratios and allocated resources for different model types across different testing cases.

Table 2: Case study of the thresholds $(h_1, h_2)$, processing ratios $(p_1, p_2, p_3)$, and allocated resources $(f_1, f_2, f_3)$ for each model type within the cascade across different testing cases. (90, 1) denotes testing on Trace 1 with an average quality requirement of 90.

|          | $h_1$ | $h_2$ | $p_1$ | $p_2$ | $p_3$ | $f_1$ | $f_2$ | $f_3$ |
|----------|-------|-------|-------|-------|-------|-------|-------|-------|
| (90, 1)  | 99    | 91    | 100%  | 94%   | 50%   | 4     | 8     | 20    |
| (85, 1)  | 74    | 64    | 100%  | 62%   | 21%   | 4     | 12    | 16    |
| (80, 1)  | 69    | 25    | 100%  | 54%   | 11%   | 6     | 14    | 12    |
| (80, 2)  | 61    | 18    | 100%  | 31%   | 3%    | 8     | 16    | 8     |
| (80, 3)  | 32    | 0     | 100%  | 23%   | 0%    | 18    | 14    | 0     |
| (70, 3)  | 10    | 0     | 100%  | 5%    | 0%    | 24    | 8     | 0     |

**Case study on parallelism strategies.** Table 3 presents a case study on parallelism strategies for each model type within the cascade across different test cases.

Table 3: Case study of the parallelism strategies for each model type within the cascade ($s_1$, $s_2$, $s_3$) across different testing cases.

| | Parallelism Strategies |
|---|---|
| (90, 1) | $s_1$: (DP=4), $s_2$: (DP=2, TP=4), $s_3$: (TP=4, PP=3), (TP=8) |
| (85, 1) | $s_1$: (DP=2, TP=2), $s_2$: (DP=6, TP=2), $s_3$: (DP=2, TP=8) |
| (80, 1) | $s_1$: (DP=6), $s_2$: (DP=5, TP=2), (TP=4), $s_3$: (TP=4, PP=3) |
| (80, 2) | $s_1$: (DP=6), (TP=2), $s_2$: (DP=8, TP=2), $s_3$: (TP=8) |
| (80, 3) | $s_1$: (DP=10), (DP=4, TP=2), $s_2$: (DP=2, TP=4), (DP=3, TP=2), $s_3$: - |
| (70, 3) | $s_1$: (DP=16), (DP=4, TP=2), $s_2$: (DP=4, TP=2), $s_3$: - |

Table 4: End-to-end throughput results evaluating CASCADIA against Sarathi-Serve.

| Trace | Ours | Sarathi-Serve | Speedup | %Improvement |
|---|---|---|---|---|
| Trace 1 | 0.2529 req/s | 0.1913 req/s | 1.322 | +32.20% |
| Trace 2 | 0.4659 req/s | 0.2385 req/s | 1.953 | +95.35% |
| Trace 3 | 0.6406 req/s | 0.3977 req/s | 1.611 | +61.08% |

## F    COMPARISON WITH SARATHI-SERVE

We evaluated Sarathi-Serve under the same experimental setup as SGLang, as described in §4.1, using traces 1–3 with an average quality requirement of 90. We used Sarathi-Serve's vLLM implementation (its most efficient variant) and tuned the chunk size to be optimal for each case. As shown in Table 4, our system achieves up to $1.95\times$ higher throughput and averages a $1.64\times$ speedup across traces.

## G    LATENCY RESULTS ON FLUCTUATING WORKLOADS

**Quantification of re-scheduling overheads.** The re-scheduling overhead consists of two components: **(i)** Algorithm runtime ($\sim$10-20s, as shown in Figure 10), and **(ii)** model reconfiguration overhead ($\sim$2-20s).

- **Re-scheduling impact on online serving.** During rescheduling, requests continue to be processed using the current deployment configuration, so there is no service interruption.

- **Reconfiguration impact on online serving.** Deployment plans typically have overlapping configurations between transitions (i.e., some model replicas retain the same deployment configuration), so these unchanged replicas can continue processing requests during reconfiguration. To further reduce the service interruption time, for replicas that do require reconfiguration, we perform **rolling updates**—reconfiguring them one at a time while others continue serving requests.

**Re-scheduling impact on baseline methods.** Note that CascadeServe also incurs similar reconfiguration overhead, while single-model baselines exhibit consistently poor performance due to lack of cascade optimization.

We further demonstrate the latency results of CASCADIA compared to CascadeServe and single-model deployment in our fluctuating workload experiments (Figure 11) with average quality requirement of 90. CASCADIA achieves 34% and 45% reduction in SLO scale for achieving 95% SLO attainment compared to CascadeServe and single-model deployment.

Table 5: Benchmarked SLO Scale for 95% SLO Attainment (Avg. Quality $\geq 90$).

| Deployment Strategy | SLO Scale | Reduction vs. CASCADIA |
|---|---|---|
| CASCADIA | 8.99 | — |
| CascadeServe | 13.55 | 34% |
| Single-Model Deployment | 16.37 | 45% |

# H    BENCHMARK GPT-4O OVERHEAD

We conducted additional experiments on H100 GPUs to benchmark single-request GPT-4o judging vs. processing latency on MT-Bench using Llama cascades (Llama3-8B $\rightarrow$ Llama3-70B). The results demonstrate that the average single-request processing latency for the small (approximately 3.05s) and large model (approximately 7.35s) is approximately 5.21s. In contrast, the single-request GPT-4o judging latency is only approximately 0.27s, as judging is prefill-bound with minimal output (1 token). This overhead is negligible compared to overall inference cost, and is already included in all experimental results (§4) reported in our paper.

# I    COMPARE WITH ROUTELLM

We conducted additional experiments comparing CASCADIA against RouteLLM with BERT-based router on Llama cascades (Llama3-8B $\rightarrow$ Llama3-70B) following the setup in Section 4.1 with average quality requirement of 80 on Traces 1 and 2. For fair comparison, we tune the deployment for each model for RouteLLM. Results show that CASCADIA achieves on average 21.3% lower SLO scale in achieving 95% SLO attainment (4.6, 2.8 vs. 5.8, 3.6) and 18.8% higher throughput (2.2, 3.5 vs. 1.9, 2.9) compared to RouteLLM.

CASCADIA's performance advantage stems from its system-algorithm co-design (§3): While RouteLLM focuses solely on routing optimization and fails to consider how system-side optimization (e.g., resource allocation, parallelism) impacts routing decisions and latency-quality trade-offs, CASCADIA jointly optimizes both aspects for better end-to-end performance.

# J    SCHEDULING OPTIMALITY

Due to the NP-hardness of the problem and the mutual dependencies between deployment and routing, providing theoretical optimality guarantees is intractable. However, we can empirically validate our approach against exhaustive search, which enumerates all feasible resource allocations, parallelism strategies, and routing thresholds, serving as an empirical optimum. Specifically, we conducted additional experiments comparing our bi-level optimization against exhaustive search on Llama cascades (Llama3-8B $\rightarrow$ Llama3-70B) following the setup in Section 4.1. To make exhaustive search computationally feasible, we applied the same deployment constraints from Section 3.2. Results show that our approach achieves near-optimal performance with only 2-6% gap compared to exhaustive search, while reducing search time from $\geq$5 minutes to 20 seconds—a $\geq$15$\times$ speedup. Notably, exhaustive search time grows exponentially with cluster size, making our bi-level approach (grows linearly) essential for practical deployment at scale.

While theoretical optimality is intractable, our method provides strong empirical performance with practical efficiency, making it suitable for real-world deployment scenarios where search overhead matters.

# K    SENSITIVITY EXPERIMENTS WITH WEAKER JUDGES

We conducted additional experiments to evaluate robustness by replacing GPT-4o with weaker judges (GPT-4o-mini and Llama3.1-70B), following the same experimental setup as Figure 7 (Llama cascade, Trace 1, quality requirement $q_{\min} = 80$).

### K.1 EXPERIMENTAL RESULTS ANALYSIS

GPT-4o-mini and Llama3.1-70B assign scores that are on average **9.4%** and **8.6%** lower than GPT-4o for the same responses, exhibiting higher variance in quality assessment. This scoring bias causes the system to route **11.1%** and **9.4%** more requests to the larger model compared to using GPT-4o. Nevertheless, Cascadia adaptively adjusts the deployment, allocating more resources to the larger model. As a result, meeting the same $q_{min}$ requires only **6.8%** and **5.5%** increase in system latency, respectively. Importantly, the system continues to satisfy the quality requirement and avoids collapse into over-routing, demonstrating that Cascadia is robust to weaker or noisier judges.

Table 6: Sensitivity to judge quality: Performance change compared to GPT-4o baseline.

| Judge | Avg Score Deviation from GPT-4o | Routing to Larger Model Increase | System Latency Increase to meet $q_{min}$ |
|---|---|---|---|
| GPT-4o (baseline) | 0% | 0% | 0% |
| GPT-4o-mini | $-9.4\%$ | $+11.1\%$ | $+6.8\%$ |
| Llama3.1-70B | $-8.6\%$ | $+9.4\%$ | $+5.5\%$ |

### K.2 JUDGE-AGNOSTIC FRAMEWORK

Our framework is **judge-agnostic**: any model capable of pairwise comparison or quality scoring can be used as the judge, including open-source models (e.g., Llama-based judges).

## L COST EFFICIENCY RESULTS

We provide a cost-efficiency analysis comparing CASCADIA against baselines. Following the experimental setup in Figure 6 with an average quality requirement of 90 ($q_{avg} = 90$), we compute the cost per request based on GPU pricing (NVIDIA H100: $2.67/hour). Results demonstrate that CASCADIA achieves **20–39%** cost reduction compared to CascadeServe and **33–61%** reduction compared to stand-alone serving.

Table 7: Cost per request (USD/req) comparison (Avg. Quality $q_{avg} = 90$, H100 GPU pricing).

| Deployment Strategy | Trace 1 | Trace 2 | Trace 3 |
|---|---|---|---|
| Stand-Alone Serving | 0.15 $/req | 0.26 $/req | 0.31 $/req |
| CascadeServe | 0.14 $/req | 0.18 $/req | 0.15 $/req |
| CASCADIA (Ours) | **0.10 $/req** | **0.11 $/req** | **0.12 $/req** |

## M DISCUSSION OF INTEGRATING PREFIX CACHING

**Additional experiments with enabling prefix caching.** In our experiments on MT-Bench (Trace 1, avg_quality=80) with the Llama cascade (Llama3-8B $\rightarrow$ Llama3-70B), enabling prefix caching changed the SLO scale required to achieve 95% SLO attainment from **4.6** to **4.5** ($\sim$2% system latency decrease, within measurement noise) and did not affect the relative gaps between CASCADIA and the baselines or the resulting scheduling decisions. Similarly, CascadeServe's SLO scale stayed the **same**, and stand-alone model serving changed from **8.2** to **8.1**.

This minor impact is reasonable due to MT-Bench's workload characteristics. MT-Bench is **decoding-heavy**, so even perfect prefix reuse would have limited impact on overall latency dominated by the decoding phase.

**Prefix caching impact on scheduling decision.** In serving scenarios where many different requests share a long, identical prefix, enabling prefix caching reduces absolute p95 latencies for all systems.

However, the relative performance gains of CASCADIA over the baselines and the resulting optimal scheduling decisions remain largely unchanged, since prefix caching **benefits all model replicas uniformly** across the cluster.

**How to incorporate prefix caching in our scheduling algorithm.** To incorporate prefix caching into our cost estimation, we can model it as a prefill reduction:

$$\text{effective\_prefill\_tokens} \approx (1 - \text{hit\_rate}) \times \text{original\_prefill\_tokens} \tag{1}$$

where hit_rate is obtained from profiling a representative subsample of the input workloads. We note that prefix caching is an orthogonal optimization technique—the bi-level scheduling methodology remains applicable and would operate on cache-adjusted latency profiles.

## N  FINE-TUNED BERT FOR CASCADING

We conducted two experiments to evaluate BERT-based judging:

1. **Realistic fine-tuning scenario.** We fine-tuned a BERT model using 70% of our experimental traces as training data and evaluated on the remaining 30% (unseen test set). The BERT judger was trained on request inputs, outputs, and quality grades across different models. We compared CASCADIA (GPT-4o) against CASCADIA (BERT) on Llama cascades (Llama3-8B → Llama3-70B) following the setup in Section 4.1 with a quality requirement of 80. Results show that CASCADIA with the BERT judger exhibits large variance in quality assessment, leading to ∼**8%** degradation in system quality (80 → 74) compared to CASCADIA (GPT-4o). This demonstrates that a less accurate judger fails to satisfy the quality requirement.

2. **Oracle fine-tuning scenario.** To isolate judging overhead from judging accuracy, we trained BERT on 100% of our experimental traces (including the test set), creating an oracle judger that perfectly replicates GPT-4o's judgments with minimal overhead. Even in this idealized scenario, CASCADIA (BERT-oracle) achieves **only <5%** better system latency than CASCADIA (GPT-4o) due to faster judging time, demonstrating that judging overhead is already negligible.

**Why LLM-as-a-Judge over BERT-based routers.** We choose LLM-as-a-Judge for two key reasons: (1) **Overhead**: As shown above, judging adds only ∼**0.27s** overhead, which is negligible compared to inference savings from routing simple requests to smaller models. (2) **Generalization**: BERT-based routers suffer from generalization problems when encountering diverse or out-of-distribution queries, whereas LLM-as-a-Judge (Zheng et al., 2023) can evaluate response quality more robustly across varied workloads and domains.

## O  THE USE OF LLMS IN WRITING

We used LLM, namely OPENAI-GPT5, to polish the writing of this manuscript. No other generative AI functionality is used in the writing of this submission.

