# OpenReview forum: "Cascadia: An Efficient Cascade Serving System for Large Language Models"
_ICLR.cc/2026/Conference — ICLR 2026 Poster_

### Official Review · Reviewer_m6H1 · 2025-11-01

**Soundness:** 3
**Presentation:** 3
**Contribution:** 3
**Rating:** 6
**Confidence:** 3

**Summary:**

The paper introduces CASCADIA, a cascade-serving system for LLMs that jointly optimizes (i) GPU/resource deployment of multiple models (including DP/TP/PP choices) and (ii) threshold-based routing across the cascade to meet a target latency–quality trade-off. It uses a bi-level optimization loop: a MILP picks per-model allocations/parallelism given a routing pattern, and a Chebyshev-guided routing solver adjusts thresholds to satisfy a user quality floor. On DeepSeek and Llama cascades, CASCADIA outperforms single-model serving and CascadeServe, reporting up to 4× tighter SLOs and 5× higher throughput, with ablations showing that both resource allocation and parallelism search are necessary.

**Strengths:**

- Real problem, 2026-relevant: serving fleets of heterogeneous LLMs under SLO and quality constraints is exactly what people are doing.
- The bi-level loop (MILP for deployment + Chebyshev for routing) is a clean way to expose the coupling between “how many requests go to 70B” and “how many GPUs the 70B should get.”
- Strong empirical numbers: up to 4× lower latency SLOs and up to 5× higher throughput than single-model; 1.7–2.5× over CascadeServe; wins also hold for Llama cascades.
- Removing parallelism search or using uniform GPU allocation degrades performance notably, so the system is not just “a nicer scheduler,” it’s actually using its degrees of freedom.
- Online re-scheduling: handling trace1 → trace2 → trace3 shifts and still beating baselines makes the system more believable for production.

**Weaknesses:**

- Routing quality is based on GPT-4o-as-a-judge. That’s fine for the paper but less fine for on-prem/air-gapped setups; we don’t see how robust the method is to weaker judges.
- Incremental vs existing cascade work: CascadeServe, AutoMix, and even 2025 routing+speculative papers are moving in the same direction.
- No cost/energy accounting, they motivate with resource efficiency and even cite energy papers, but do not report $/req or some energy unit /req, that’s what operators would care about.
- If the quality distributions used in the routing solver are mis-estimated (domain shift, very hard math/coding workloads), the Chebyshev objective could over-route to large models and wipe out the latency gains.

**Questions:**

1. The routing quality depends on GPT-4o as the judge. If we swap in a weaker/open-source judge (e.g. Llama-3-70B with an arena prompt), does CASCADIA still satisfy the same 𝑞 min, or does it start over-routing to bigger models? A small sensitivity table would help.
2. Rescheduling policy. How exactly do you detect a “significant” workload shift before re-running the scheduler, and do you use any hysteresis to avoid oscillating between plans?
3. Can the deployment MILP be warm-started from the previous solution to keep solve time low when only the routing thresholds change slightly?

---

> ### Author Response · Authors · 2025-11-20
>
> > W1. & Q1. Routing quality is based on GPT-4o-as-a-judge. That’s fine for the paper but less fine for on-prem/air-gapped setups; we don’t see how robust the method is to weaker judges. The routing quality depends on GPT-4o as the judge. If we swap in a weaker/open-source judge (e.g. Llama-3-70B with an arena prompt), does CASCADIA still satisfy the same 𝑞 min, or does it start over-routing to bigger models? A small sensitivity table would help.
>
> **Sensitivity experiments with weaker judges.** We conducted additional experiments to evaluate robustness by replacing GPT-4o with weaker judges (GPT-4o-mini and Llama3.1-70B), following the same experimental setup as Figure 7 (Llama cascade, Trace 1, quality requirement q_min = 80).
>
> **Experimental results analysis.** GPT-4o-mini and Llama3.1-70B assign scores that are on average **9.4%** and **8.6%** lower than GPT-4o for the same responses, exhibiting higher variance in quality assessment. This scoring bias causes the system to route **11.1%** and **9.4%** more requests to the larger model compared to using GPT-4o. Nevertheless, Cascaida adaptively adjusts the deployment, allocating more resources to the larger model. As a result, meeting the same $q_{min}$ requires only **6.8%** and **5.5%** increase in system latency, respectively. Importantly, the system continues to satisfy the quality requirement and avoids collapse into over-routing, demonstrating that Cascadia is robust to weaker or noisier judges.
>
> | Judge | Avg Score Deviation from GPT-4o | Routing to Larger Model | System Latency |
> |-------|----------------------------|------------------|------------------|
> | GPT-4o (baseline) | 0% | — | 0% |
> | GPT-4o-mini | -9.4% | +11.1% | +6.8% |
> | Llama3.1-70B | -8.6% | +9.4% | +5.5% |
>
> **Judge-agnostic framework.** Our framework is **judge-agnostic**: any model capable of pairwise comparison or quality scoring can be used as the judge, including open-source models (e.g., Llama-based judges) deployed in on-prem or air-gapped environments.
>
> Thank you for your valuable insight; we have added this discussion to the paper (see footnote in page 6 and Appendix K in the revised manuscript).
>
>
> > W2. Incremental vs existing cascade work: CascadeServe, AutoMix, and even 2025 routing+speculative papers are moving in the same direction.
>
> **Compare with CascadeServe.** As discussed in Section 2, CascadeServe overlooks key LLM-specific workload characteristics such as input/output sequence length distributions and request complexity. In contrast, Cascadia jointly considers workload heterogeneity, resource allocation, parallelism strategies, and routing decisions. Experimental results demonstrate that Cascadia achieves up to **2.5×** lower latency and **3.3×** higher throughput compared to CascadeServe (Figures 5-6).
>
> **Comparison with AutoMix.** AutoMix is primarily focused on routing policy design—it uses self-verification and a POMDP-based router to decide when to escalate from smaller to larger models. This is **orthogonal** to our main focus: system-level optimization for cascade serving。
>
>
> > W3. No cost/energy accounting, they motivate with resource efficiency and even cite energy papers, but do not report $/req or some energy unit /req, that’s what operators would care about.
>
> **Cost efficiency results.** We provide cost-efficiency analysis comparing Cascadia against baselines. Following the experimental setup in Figure 6 with an average quality requirement of 90, we compute the cost per request based on GPU pricing (NVIDIA H100: $2.67/hour). Results demonstrate that Cascadia achieves **20-39%** cost reduction compared to CascadeServe and **33-61%** reduction compared to stand-alone serving:
>
> |              | Trace 1  | Trace 2  | Trace 3  |
> |--------------|-----------------|-----------------|-----------------|
> | Stand-Alone  | 0.15  $/req          | 0.26  $/req          | 0.31 $/req           |
> | CascadeServe | 0.14   $/req         | 0.18   $/req         | 0.15 $/req           |
> | Cascadia     | **0.10**  $/req          | **0.11**     $/req       | **0.12**     $/req       |
>
> Thank you for your insight; we have added this additional experiment to the paper (see Section 4.2 in page 8 and Appendix L in the revised manuscript).

---

> ### Author Response · Authors · 2025-11-20
>
> > W4. If the quality distributions used in the routing solver are mis-estimated (domain shift, very hard math/coding workloads), the Chebyshev objective could over-route to large models and wipe out the latency gains.
>
> Domain shift or distribution mis-estimation could potentially affect routing decisions. However, our system includes mechanisms to mitigate this risk:
>
> **Re-scheduling mechanism.** Our re-scheduling mechanism (Section 4.4) monitors real-time workload characteristics and evaluates whether re-scheduling is necessary every 5 minutes. This adaptive approach prevents sustained over-routing caused by domain shift or distribution mis-estimation.
>
> **Additional experiments on robustness.** We conducted additional experiments with intentionally mis-estimated quality distributions (15% lower mean quality scores than actual) on Trace 1 with Llama cascades (Llama3-8B → Llama3-70B) and quality requirement of 80 (the same setup as Figure 7). Results show that Cascadia's performance degrades gracefully: when quality is underestimated by 15% (pessimistic case causing over-routing to larger models), p95 latency increases by only **~12%** while still maintaining **1.57×** improvement over baselines.
>
>
> > Q2. Rescheduling policy. How exactly do you detect a “significant” workload shift before re-running the scheduler, and do you use any hysteresis to avoid oscillating between plans?
>
> **Rescheduling policy and workload shift detection.** Our system re-runs the scheduling algorithm every 5 minutes using recent workload statistics. To avoid oscillating between plans, we employ a hysteresis mechanism: we switch to a new deployment plan only when the estimated cost savings over the next 5 minutes exceed the switching cost by a factor of **ε = 1.5** (by default). The switching cost accounts for resource re-allocation and parallelism reconfiguration. This threshold prevents frequent changes from minor fluctuations while enabling adaptation to significant shifts.
>
>
> > Q3. Can the deployment MILP be warm-started from the previous solution to keep solve time low when only the routing thresholds change slightly?
>
>
> **Warm-starting the deployment MILP.** Yes, the deployment MILP can be warm-started from the previous solution to reduce solve time. We initialize the MILP solver with **the previous solution as a starting point**, which allows the solver to verify optimality or find nearby solutions more quickly.

---

### Official Review · Reviewer_jBdC · 2025-11-01

**Soundness:** 3
**Presentation:** 3
**Contribution:** 3
**Rating:** 6
**Confidence:** 3

**Summary:**

This paper proposes Cascadia, a novel cascade serving framework for efficient & effective LLM serving. Cascadia employs a bi-level optimization strategy to compute the optimal serving strategy which includes the deployment strategy (i.e., the resource allocation and parallelism strategies) as well as the routing strategy (i.e., which user requests should be consumed by which models). Extensive experiments on real-world datasets demonstrate the effectiveness of the proposed approach.

**Strengths:**

1. This paper studies efficient and effective LLM serving, which is critical problem for a wide range of real-world applications.
2. This paper introduces Cascadia with rigorous and solid technical developments.
3. Cascadia achieves up to 4x lower latency deadlines and 5x higher system throughputs, which are impressive.

**Weaknesses:**

1. In Algorithm 1, Cascadia relies on iteratively optimizing both deployment and routing strategies. It remains unclear if the bi-level optimization is theoretically optimal or not. Such guarantees could be critical in practical scenarios.
2. LLM routing is another well-studied technique aiming for efficient & effective LLM serving, which is under-discussed in this paper. Authors may want to discuss and compare to this line of work to better position the contribution of this paper. Several example references are as follows,

[1] Ong, Isaac, et al. "RouteLLM: Learning to Route LLMs from Preference Data." The Thirteenth International Conference on Learning Representations.
[2] Ding, Dujian, et al. "BEST-Route: Adaptive LLM Routing with Test-Time Optimal Compute." Forty-second International Conference on Machine Learning.

**Questions:**

What is the typical/expected number of iterations required to achieve stable solutions? If the number of iteration tends to be huge, it can lead to non-negligible overheads and compromise the efficiency gains.

---

> ### Author Response · Authors · 2025-11-20
>
> > W1. In Algorithm 1, Cascadia relies on iteratively optimizing both deployment and routing strategies. It remains unclear if the bi-level optimization is theoretically optimal or not. Such guarantees could be critical in practical scenarios.
>
> **Theoretical optimality.** Due to the NP-hardness of the problem and the mutual dependencies between deployment and routing, providing theoretical optimality guarantees is intractable. However, we can empirically validate our approach against **exhaustive search**, which enumerates all feasible resource allocations, parallelism strategies, and routing thresholds, serving as an empirical optimum.
>
> Specifically, we conducted additional experiments comparing our bi-level optimization against exhaustive search on Llama cascades (Llama3-8B → Llama3-70B) following the setup in Section 4.1. To make exhaustive search computationally feasible, we applied the same deployment constraints from Section 3.2. Results show that our approach achieves near-optimal performance with only **2-6%** gap compared to exhaustive search, while reducing search time from >5 minutes to ~20 seconds—a **>15×** speedup. Notably, exhaustive search time grows **exponentially** with cluster size, making our bi-level approach (grows **linearly**) essential for practical deployment at scale.
>
> **Practical advantages.** While theoretical optimality is intractable, our method provides strong empirical performance with practical efficiency, making it suitable for real-world deployment scenarios where search overhead matters.
>
> Thank you for mentioning this; we have added this discussion to the paper (see Appendix J in the revised manuscript).
>
> > W2. LLM routing is another well-studied technique aiming for efficient & effective LLM serving, which is under-discussed in this paper. Authors may want to discuss and compare to this line of work to better position the contribution of this paper. Several example references are as follows,
>
> **Additional experiments with RouteLLM.** We conducted additional experiments comparing Cascadia against RouteLLM (https://github.com/lm-sys/RouteLLM) on Llama cascades (Llama3-8B → Llama3-70B) following the setup in Section 4.1 with average quality requirement of 80 on Traces 1 and 2. For fair comparison, we tune the deployment for each model for RouteLLM. Results show that Cascadia achieves on average **21.3%** lower SLO scale in achieving 95% SLO attainment (4.6, 2.8 vs. 5.8, 3.6) and **18.8%** higher throughput (2.2, 3.5 vs. 1.9, 2.9) compared to RouteLLM.
>
> Cascadia's performance advantage stems from its **system-algorithm co-design** (Section 3): While RouteLLM focuses solely on routing optimization and fails to consider how system-side optimization (e.g., resource allocation, parallelism) impacts routing decisions and latency-quality trade-offs, Cascadia jointly optimizes both aspects for better end-to-end performance.
>
> Thank you for your valuable suggestion; we have added this additional experiment to the paper (see Section 4.2 in page 8 and Appendix I in the revised manuscript).
>
> > Q1. What is the typical/expected number of iterations required to achieve stable solutions? If the number of iteration tends to be huge, it can lead to non-negligible overheads and compromise the efficiency gains.
>
> **Scheduling overhead.** The typical number of iterations required to achieve stable solutions is **~40 iterations**. We present a detailed analysis of the total scheduling overhead in Section 4.4 (Figure 10). In our setup (32 GPUs), scheduling completes within **20s**. For larger clusters (e.g., 80 GPUs), it finishes within **one minute**. These results demonstrate the algorithm’s efficiency and scalability across test cases and cluster sizes.

---

### Official Review · Reviewer_11UR · 2025-11-03

**Soundness:** 2
**Presentation:** 3
**Contribution:** 2
**Rating:** 4
**Confidence:** 4

**Summary:**

This paper introduces CASCADIA, a novel cascade serving framework designed explicitly to schedule request routing and deploy model cascades for fast, quality-preserving LLM serving. CASCADIA employs a bi-level optimization method: at the deployment level, it uses a mixedinteger linear program to select resource allocations and parallelism strategies based on LLM information and workload characteristics; at the routing level, it applies a Chebyshev-guided method to iteratively co-optimize the routing strategy and the system deployment produced by the deployment level.

It uses mixed-integer linear programming (MILP) to determine the optimal deployment plan given a routing strategy. It balances response latency and output quality. Within each cascade stage, CASCADIA supports various parallelism strategies (e.g., tensor and pipeline parallelism), which allows it to automatically select the optimal strategy based on model size, incoming workload, and routing decisions.

**Strengths:**

Stengths:

1. Efficient serving multiple LLM to balance accuracy and latency is an important topic.

2. The proposed cascading method intuitively can help the multi-model serving system.

3. Extensive experiments show the performance.

**Weaknesses:**

1. The main concern is on the real-time efficiency and cost. LLM serving is an online process. If using GPT-4 to judge the small model response, runs GPT-4 takes a few seconds and the cost is expensive.

2. Time to first token is also very long. For simple prompt, it also needs to wait until GPT-4 finishes the judge.

3. The baselines are insufficient. BERT-based router [1, 2, 3] that directly routes prompt to multiple LLMs can be compared.


[1] https://github.com/vllm-project/semantic-router

[2] Tensoropera router: A multi-model router for efficient llm inference

[3] RouteLLM: Learning to Route LLMs with Preference Data

**Questions:**

1. Maybe instead of using GPT-4, fine tune a BERT for cascading?

---

> ### Author Response · Authors · 2025-11-20
>
> > W1. The main concern is on the real-time efficiency and cost. LLM serving is an online process. If using GPT-4 to judge the small model response, runs GPT-4 takes a few seconds and the cost is expensive.
>
> **Benchmark judging vs. processing latency to demonstrate GPT-4o overhead is negligible.** We conducted additional experiments on H100 GPUs to benchmark single-request GPT-4o judging vs. processing latency on MT-Bench using Llama cascades (Llama3-8B → Llama3-70B). The results demonstrate that the average single-request processing latency for the small (on average 3.05s) and large model (on average 7.35s) is on average **5.21s**. In contrast, the single-request GPT-4o judging latency is only **~0.27s**, as judging is prefill-bound with minimal output (1 token). This overhead is negligible compared to overall inference cost, and is already included in all experimental results (Section 4) reported in our paper.
>
> Thank you for mentioning this; we have made further clarification about this overhead in the paper (see footnote on page 6 in the revised manuscript).
>
>
> **Why LLM-as-a-Judge over BERT-based routers.** We choose LLM-as-a-Judge for two key reasons: (1) **Overhead**: As shown above, judging adds only **~0.27s** overhead, which is negligible compared to inference savings from routing simple requests to smaller models. (2) **Generalization**: BERT-based routers suffer from generalization problems when encountering diverse or out-of-distribution queries, whereas LLM-as-a-Judge (NIPS'23) can evaluate response quality more robustly across varied workloads and domains.
>
>
> > Q1. Maybe instead of using GPT-4, fine tune a BERT for cascading?
>
>
> **Additional experiments using fine-tuned BERT for cascading.** We conducted two experiments to evaluate BERT-based judging:
>
> (1) **Realistic fine-tuning scenario.** We fine-tuned a BERT model using 70% of our experimental traces as training data and evaluated on the remaining 30% (unseen test set). The BERT judger was trained on request inputs, outputs, and quality grades across different models. We compared Cascadia (GPT-4o) against Cascadia (BERT) on Llama cascades (Llama3-8B → Llama3-70B) following the setup in Section 4.1 with a quality requirement of 80. Results show that Cascadia with the BERT judger exhibits large variance in quality assessment, leading to **~8%** degradation in system quality (80 → 74) compared to Cascadia (GPT-4o). This demonstrates that a less accurate judger fails to satisfy the quality requirement.
>
> (2) **Oracle fine-tuning scenario.** To isolate judging overhead from judging accuracy, we trained BERT on 100% of our experimental traces (including the test set), creating an oracle judger that perfectly replicates GPT-4o's judgments with minimal overhead. Even in this idealized scenario, Cascadia (BERT-oracle) achieves **only <5%** better system latency than Cascadia (GPT-4o) due to faster judging time, demonstrating that judging overhead is already negligible.
>
> **Clarification on the scope of our contribution.** The main focus of this paper is **algorithm-system co-design** for cascading systems. As long as the judger does not incur significant overhead, the choice of judger (BERT vs. GPT-4o) is **orthogonal** to our core contribution. Our framework is judger-agnostic and can incorporate any quality evaluation mechanism.
>
>
> Thank you for your suggestion; we have added this additional experiment and discussion to the paper (Appendix N in the revised manuscript).
>
> > W2. Time to first token is also very long. For simple prompt, it also needs to wait until GPT-4 finishes the judge.
>
>
> **Clarification on TTFT in cascade systems.** TTFT (Time To First Token) measures when the user receives the first output token, confirming that their request has been scheduled for serving. Cascade systems are able to maintain TTFT comparable to single-model serving (using only the smallest model) through **streaming-based** implementation: tokens from the first-stage model are **streamed immediately** to the user, while judging and escalation proceed concurrently in the background. If the judger determines that escalation is necessary (i.e., the request should be further processed by a larger model), the system seamlessly replaces the initial response with streaming output from the larger model. This design ensures that TTFT remains equivalent to single-model deployment while preserving the latency and quality benefits of cascade serving.

---

> ### Author Response · Authors · 2025-11-20
>
> > W3. The baselines are insufficient. BERT-based router that directly routes prompt to multiple LLMs can be compared.
>
> **Additional experiments with RouteLLM.** We conducted additional experiments comparing Cascadia against RouteLLM with BERT-based router (https://github.com/lm-sys/RouteLLM) on Llama cascades (Llama3-8B → Llama3-70B) following the setup in Section 4.1 with average quality requirement of 80 on Traces 1 and 2. For fair comparison, we tune the deployment for each model for RouteLLM. Results show that Cascadia achieves on average **21.3%** lower SLO scale in achieving 95% SLO attainment (4.6, 2.8 vs. 5.8, 3.6) and **18.8%** higher throughput (2.2, 3.5 vs. 1.9, 2.9) compared to RouteLLM.
>
> Cascadia's performance advantage stems from its **system-algorithm co-design** (Section 3): While RouteLLM focuses solely on routing optimization and fails to consider how system-side optimization (e.g., resource allocation, parallelism) impacts routing decisions and latency-quality trade-offs, Cascadia jointly optimizes both aspects for better end-to-end performance.
>
>
> Thank you for your valuable suggestion; we have added this additional experiment to the paper (see Section 4.2 in page 8 and Appendix I in the revised manuscript).

---

### Official Review · Reviewer_HDzY · 2025-11-03

**Soundness:** 3
**Presentation:** 3
**Contribution:** 3
**Rating:** 6
**Confidence:** 4

**Summary:**

The paper presents CASCADIA, an efficient cascade serving system for large language models that jointly optimizes resource allocation and request routing across hierarchies of model sizes, enabling fast, high-quality, and cost-effective LLM inference. By formulating cascade serving as a bi-level optimization problem and dynamically adapting deployment strategies, CASCADIA achieves significantly lower latency and higher throughput compared to single-model and existing cascade baselines, while maintaining answer quality across diverse workloads.

**Strengths:**

1. The joint formulation of resource allocation and adaptive inference via model cascades is a novel approach to cost-efficient LLM inference and is a very important problem that needs to be solved before cascades can be deployed in real world systems. The paper also considers heterogeneity in model and workload characteristics which are important considerations in real settings.

2. The paper proposes a viable solution to the problem via bi-level optimization that helps to find an appropriate deployment and routing strategy and shows clear improvements over baselines in experiments

**Weaknesses:**

1. Some of the details of the approach are not clearly explained. I have added several questions below around points that were not clear to me.

2. While the bi-level optimization itself doesn't seem to be taking too long to solve in online settings (Section 4.4), the latency of re-allocating the models/changing the parallelization may be high.

3. The approach does not consider prefix caching even though the traces used in the experiments do contain multi-turn conversations and prefix caching in such settings and may affect the latencies of both the baselines and Cascadia. Even if it is difficult to incorporate prefix caching in the optimization formulation, I believe when running the traces prefix caching should be enabled to see if it causes the results to deviate significantly from what is expected after solving the optimization problem.

**Questions:**

1. Why do you try to minimize the maximum latency across models, L, in the MILP, when the latency of a query will be the sum of the latencies of the models that it passes through?

2. Why do you consider separate thresholds for each model when determining the routing strategy even though the same input is passed through the models until a satisfactory output is obtained (if the input is the same then the threshold on the output score should also be the same for all models)?

3. $L(\theta)$ in line 263 appears to be non-differentiable. If that is indeed the case, please clarify how $\theta$ is updated to converge to the minima of the optimization problem.

4. Please provide a citation for the baseline CascadeServe in Section 4.

5. How is query complexity measured when looking for a shift in workload characteristics in Section 4.4?

6. How does CascadeServe handle distribution shifts? Does it not make any changes under distribution shift?

7. Can you quantify the scheduling overhead (line 476) in terms of additional latency (and not just throughput) when re-scheduling under online workloads? For e.g. if there is a spike in the P95 latency during the rescheduling window then that would not be a good thing.

---

> ### Author Response · Authors · 2025-11-20
>
> > W2. While the bi-level optimization itself doesn't seem to be taking too long to solve in online settings (Section 4.4), the latency of re-allocating the models/changing the parallelization may be high.
>
>
> **Re-allocation overhead can be minimized.** Re-allocation overhead can be minimized to **<20s** for the largest model through practical optimizations: pre-loading model weights in CPU memory, leveraging fast host-to-device transfer (e.g., PCIe 5.0 x16 with more than 30 GB/s host-to-device bandwidth, which is generally available on Hopper-generation GPUs), and pre-computing CUDA graphs for each parallelism configuration. Assuming re-allocation triggers every 10 minutes (as our testing case in Figure 11), this represents only **~3%** overhead.
>
>
> > W3. The approach does not consider prefix caching even though the traces used in the experiments do contain multi-turn conversations and prefix caching in such settings and may affect the latencies of both the baselines and Cascadia. Even if it is difficult to incorporate prefix caching in the optimization formulation, I believe when running the traces prefix caching should be enabled to see if it causes the results to deviate significantly from what is expected after solving the optimization problem.
>
>
> **Additional experiments with enabling prefix caching.** In our experiments on MT-Bench (Trace 1, avg_quality=80) with the Llama cascade (Llama3-8B → Llama3-70B), enabling prefix caching changed the SLO scale required to achieve 95% SLO attainment from **4.6** to **4.5** (~2% system latency decrease, within measurement noise) and did not affect the relative gaps between Cascadia and the baselines or the resulting scheduling decisions. Similarly, CascadeServe's SLO scale stayed the **same**, and stand-alone model serving changed from **8.2** to **8.1**.
>
>
> This minor impact is reasonable due to MT-Bench's workload characteristics. MT-Bench is **decoding-heavy**, so even perfect prefix reuse would have limited impact on overall latency dominated by the decoding phase.
>
>
> **Prefix caching impact on scheduling decision.** In serving scenarios where many different requests share a long, identical prefix, enabling prefix caching reduces absolute p95 latencies for all systems. However, the relative performance gains of Cascadia over the baselines and the resulting optimal scheduling decisions remain largely unchanged, since prefix caching **benefits all model replicas uniformly** across the cluster.
>
> **How to incorporate prefix caching in our scheduling algorithm.** To incorporate prefix caching into our cost estimation, we can model it as a prefill reduction:
>
> `effective_prefill_tokens ≈ (1 - hit_rate) × original_prefill_tokens`
>
> where `hit_rate` is obtained from profiling a representative subsample of the input workloads. We note that prefix caching is an orthogonal optimization technique—the bi-level scheduling methodology remains applicable and would operate on cache-adjusted latency profiles.
>
> Thank you for your insight; we have added this additional discussion and experiment to the paper (see Appendix M in the revised manuscript).
>
> > Q1. Why do you try to minimize the maximum latency across models, L, in the MILP, when the latency of a query will be the sum of the latencies of the models that it passes through?
>
> Thank you for this valuable question. We clarify why our min-max formulation correctly optimizes per-request latency:
>
> **Per-request latency.** For any individual request, its end-to-end latency equals the sum of processing times across all models it passes through (e.g., if a request goes through $c_1$ and $c_2$, its latency = $t_1$ + $t_2$, where $t_1$ and $t_2$ are the processing times at each model).
>
> **Key insight: Independent workload optimization.** Given the routing strategy from the routing solver (Section 3.3), the deployment solver knows **the workload distribution**—how many models that each subsampled request will go through. Crucially, each model type processes its respective workload independently:
>
> - Model $c_1$ processes 100% of requests with p95 latency $L_1$
> - Model $c_2$ processes X% of requests with p95 latency $L_2$
> - Model $c_3$ processes Y% of requests with p95 latency $L_3$
>
> Given these workloads for all model types, the latencies {$L_1$, $L_2$, $L_3$} are independent optimization problems. Therefore, **separately minimizing each model's p95 latency is equivalent to minimizing the system's overall p95 latency**.

---

> ### Author Response · Authors · 2025-11-20
>
> > Q2. Why do you consider separate thresholds for each model when determining the routing strategy even though the same input is passed through the models until a satisfactory output is obtained (if the input is the same then the threshold on the output score should also be the same for all models)?
>
> **Why model-specific thresholds.** The latency–quality profiles of different models are not the same. Because our objective is to maintain the **system-level** response quality while minimizing latency, using model-specific thresholds allows the scheduler to finely control how many requests each model processes, balancing **(i)** the heterogeneous system latency of different model types and **(ii)** their heterogeneous quality gains. A single global threshold, however, forces all models to accept/reject requests at the same confidence level regardless of their capacity differences, which is suboptimal.
>
> **Additional experiments with uniform thresholds.** To validate this, we conducted an ablation on Llama cascades (Llama3-8B → Llama3-70B, Trace 1, quality=70) enforcing uniform thresholds, following the experimental setup in Section 4.1. The uniform-threshold variant exhibits **13%** higher system latency in achieving 95% SLO attainment compared to Cascadia's per-model thresholds (2.39 vs. 2.12 in SLO scale).
>
> > Q3. $L(\theta)$ in line 263 appears to be non-differentiable. If that is indeed the case, please clarify how $\theta$ is updated to converge to the minima of the optimization problem.
>
> Yes, $L(\theta)$ is non-differentiable. However, it is convergeable since $\theta$ is within a **finite discrete set**.
>
> To be specific, the judger outputs integer quality scores in the discrete range [0, 100], so each threshold $h_i$ in $\theta$ is selected from the same discrete domain (see Table 2 in Appendix E for examples: thresholds 99, 91, 74, etc.). Consequently, we perform grid search to solve `min over θ ∈ Θ: J(θ)` over a finite discrete set. This ensures convergence to the global minimum within the candidate set, requiring no gradient information.
>
>
> > Q4. Please provide a citation for the baseline CascadeServe in Section 4.
>
> Thank you for catching this. We have added the citation in §4.1 for clarity in the updated manuscript.
>
> > Q5. How is query complexity measured when looking for a shift in workload characteristics in Section 4.4?
>
> **Query complexity is measured by subsampling.** The query complexity is measured by subsampling 5% of incoming requests, routing them through all model types, and monitoring the quality score distributions from these sampled requests. The experimental results of Cascadia include such overhead.
>
> Thank you for mentioning this; we have clarified it in the paper (see footnote in page 9 in the revised manuscript).
>
>
> > Q6. How does CascadeServe handle distribution shifts? Does it not make any changes under distribution shift?
>
> CascadeServe (Arxiv'24) handles distribution shifts by monitoring request arrival rates and adjusting model deployment accordingly. However, it only considers arrival rate changes and fails to account for other LLM-specific workload characteristics that Cascadia considers, such as input/output length distributions and request complexity, resulting in degraded performance.

---

> ### Author Response · Authors · 2025-11-20
>
> > Q7. Can you quantify the scheduling overhead (line 476) in terms of additional latency (and not just throughput) when re-scheduling under online workloads? For e.g. if there is a spike in the P95 latency during the rescheduling window then that would not be a good thing.
>
>
> **Quantification of re-scheduling overheads.** The re-scheduling overhead consists of two components: **(i)** Algorithm runtime (10-20s, as shown in Figure 10), and **(ii)** model reconfiguration overhead (~2-20s).
>
>
> - **Re-scheduling impact on online serving.** During rescheduling, requests continue to be processed using the current deployment configuration, so there is no service interruption.
> - **Reconfiguration impact on online serving.** Deployment plans typically have overlapping configurations between transitions (i.e., some model replicas retain the same deployment configuration), so these unchanged replicas can continue processing requests during reconfiguration. To further reduce the service interruption time, for replicas that do require reconfiguration, we perform **rolling updates**—reconfiguring them one at a time while others continue serving requests.
>
> **Re-scheduling impact on baseline methods**. Note that CascadeServe also incurs similar reconfiguration overhead, while single-model baselines exhibit consistently poor performance due to lack of cascade optimization.
>
> **Additional benchmarked latency results.** We further demonstrate the latency results of Cascadia compared to CascadeServe and single-model deployment in our fluctuating workload experiments (Figure 11) with average quality requirement of 90. Cascadia achieves **34%** and **45%** reduction in SLO scale for achieving 95% SLO attainment compared to CascadeServe and single-model deployment (8.99 vs. 13.55 vs. 16.37).
>
> Thank you for your insight; we have added this additional discussion and experiment to the paper (see Section 4.4 and Appendix G in the revised manuscript).

---

### Author Response · Authors · 2025-12-02

Dear Area Chair,

We first appreciate your time and effort in handling our submission.

As the discussion period is coming to an end, we would like to provide some summarized context to ensure you have as much relevant information as possible when making the final decision.

> ### **Summary of Our Main Contribution**

Existing cascade-based serving systems fail to address three critical challenges:
- The heterogeneous resource requirements of different LLMs.
- The varying characteristics of incoming workloads.
- The need for joint optimization between system deployment and routing strategies.

Cascadia addresses these challenges through a **bi-level optimization approach** that co-optimizes resource allocation, parallelism strategy, and routing strategy. This system-algorithm co-design achieves up to 4$\times$ tighter latency SLOs and 5$\times$ higher throughput compared to both single-model and state-of-the-art cascade serving systems while maintaining target answer quality.


> ### **Regarding Paper Reviews**

**Three reviewers (HDzY, jBdC, m6H1)** rated soundness, presentation, and contribution as **good** (scores of 3/3/3) and gave overall ratings of **6**. All of them acknowledged the **significance and novelty** of our work. Their primary suggestions centered on:

- Additional experiments: E.g., prefix caching evaluation, scheduling optimality analysis, cost efficiency analysis, latency results under fluctuating workloads, and ablation studies with uniform thresholds.
- Further clarifications: E.g., query complexity measurement, judging overhead, re-scheduling mechanism details.

Our comprehensive responses—comprising several new experiments detailed in the rebuttal and Appendices F-N, along with multiple clarifications—systematically responded to all their concerns.

**Reviewer 11UR** gave an overall rating of **4** and acknowledged the **importance** of our work. The reviewer raised concerns about judging overhead, time-to-first-token (TTFT) latency, and insufficient baseline comparisons. We fully responded to these concerns by:

- Additional experiments: Benchmarking GPT-4o judging overhead, baseline comparisons with RouteLLM, and BERT-based judging experiments across realistic and oracle scenarios.
- Further clarifications: Streaming-based implementation for TTFT.

We believe our comprehensive responses address the reviewer's concerns.



> ### **Revisions in the Updated Manuscript**

We added the following changes to our updated manuscript based on the reviewers' valuable suggestions:

*Additional Experiments:*
- Prefix caching evaluation (Appendix M) – Reviewer HDzY;
- Comparison with RouteLLM showing 21.3% lower SLO scale and 18.8% higher throughput (Section 4.2, Appendix I) – Reviewers 11UR, jBdC;
- Sensitivity analysis with weaker judges (GPT-4o-mini, Llama3.1-70B) (Appendix K) – Reviewer m6H1
- Fine-tuned BERT for cascading experiments (Appendix N) – Reviewer 11UR;
- Scheduling optimality analysis against exhaustive search (Appendix J) – Reviewer jBdC;
- Cost efficiency analysis showing 20-39% cost reduction vs. CascadeServe (Section 4.2, Appendix L) – Reviewer m6H1;
- Latency results under fluctuating workloads (Section 4.4, Appendix G) – Reviewer HDzY;
- GPT-4o judging overhead benchmarks (Appendix H) – Reviewer 11UR.

*Clarifications:*
- Query complexity measurement methodology (footnote on page 9) – Reviewer HDzY;
- Judging overhead (footnote on page 6) – Reviewers 11UR, m6H1;
- CascadeServe citation (Section 4.1) – Reviewer HDzY;
- Re-scheduling mechanism details (Section 4.4, Appendix G) – Reviewers HDzY, m6H1.

We believe these revisions significantly strengthen the paper and comprehensively respond to the core concerns raised by the reviewers.

Best regards,
The Authors of Submission 12568

---

### Meta-Review · Area_Chair_EVUC · 2026-01-12

**Summary:**

It received scores of 6,4,6,6.

The reviewers point out several strengths including novel joint formulation of resource allocation and adaptive inference for cost-efficient LLM cascades; focuses on a real, timely, and practically important problem in multi-LLM serving; principled bi-level optimization that captures deployment-routing coupling; supports heterogeneous models and workloads; intuitive cascading design for balancing accuracy and latency; strong experimental results with large gains in latency and throughput; robustness demonstrated across models and workload shifts.

They also raise several weaknesses pointed out below that are mostly addressed by the authors.

**Reviewer Concerns:**

The main concerns are: practical latency and online serving concerns; ignoring prefix caching effects; real-time efficiency and cost of gpt-4-based judging; insufficient baseline comparisons; limited theoretical guarantees of the optimization framework; under-positioning relative to prior llm routing and cascade work; missing cost and energy metrics. Authors provided response that addressed most of the concerns.

**Reviewer Scores:**

Given the above points, I believe reviewers would not have reduced their rating if got the chance to reply.

---

### Decision · Program_Chairs · 2026-01-26

Accept (Poster)